# Ciliary transcription factors and miRNAs precisely regulate Cp110 levels required for ciliary adhesions and ciliogenesis

Peter Walentek[1]*, Ian K Quigley[2], Dingyuan I Sun[1], Umeet K Sajjan[1], Christopher Kintner[2], Richard M Harland[1]*

[1]Division of Genetics, Genomics and Development, Center for Integrative Genomics, Department of Molecular and Cell Biology, University of California, Berkeley, United States; [2]Molecular Neurobiology Laboratory, Salk Institute for Biological Studies, La Jolla, United States

**Abstract** Upon cell cycle exit, centriole-to-basal body transition facilitates cilia formation. The centriolar protein Cp110 is a regulator of this process and cilia inhibitor, but its positive roles in ciliogenesis remain poorly understood. Using *Xenopus* we show that Cp110 inhibits cilia formation at high levels, while optimal levels promote ciliogenesis. Cp110 localizes to cilia-forming basal bodies and rootlets, and is required for ciliary adhesion complexes that facilitate Actin interactions. The opposing roles of Cp110 in ciliation are generated in part by coiled-coil domains that mediate preferential binding to centrioles over rootlets. Because of its dual role in ciliogenesis, Cp110 levels must be precisely controlled. In multiciliated cells, this is achieved by both transcriptional and post-transcriptional regulation through ciliary transcription factors and microRNAs, which activate and repress *cp110* to produce optimal Cp110 levels during ciliogenesis. Our data provide novel insights into how Cp110 and its regulation contribute to development and cell function.

*For correspondence: walentek@ berkeley.edu (PW); harland@ berkeley.edu (RMH)

**Competing interests:** The authors declare that no competing interests exist.

## Introduction

Cilia are membrane-covered cell protrusions containing an axoneme of microtubules. Modified centrioles, called basal bodies, dock to the cell membrane, serve as microtubule organizing centers (MTOCs) during cilia formation and anchor cilia to the membrane as well as to the Actin cytoskeleton (*Marshall, 2008*). Because centrioles also act as MTOCs during spindle formation, cell division and cilia formation are mutually exclusive events. Thus, switching from cell division to cilia formation needs precise molecular regulation at the centriole (*Avidor-Reiss and Gopalakrishnan, 2013*). One key event during this process is the removal of the Centriolar Coiled Coil Protein 110kDa (Cp110) from the distal end of the mother centriole, which then matures into a basal body (*Tsang and Dynlacht, 2013*). Failure of distal end removal or excess cellular levels of Cp110 prevent cilia formation in various cell types. Conversely, loss of Cp110 was suggested to initiate aberrant cilia formation during the cell cycle (*Spektor et al., 2007*). Of note, some studies indicate that *Cp110* knockdown initiates elongation of cytoplasmic centrioles, rather than *bona fide* cilia formation (*Schmidt et al., 2009*).

In our previous work, we demonstrated that Cp110 also inhibits cilia formation in multi-ciliated cells (MCCs) of mucociliary epithelia (*Song et al., 2014*). MCCs can form >100 basal bodies, and their biogenesis occurs through an alternative, MCC-specific deuterosome pathway (*Brooks and Wallingford, 2014*; *Zhang and Mitchell, 2015*). MCC cilia are motile and account for the generation of directional extracellular fluid flow along epithelia, such as that required for mucus clearance from the conducting airways (*Mall, 2008*). Interestingly, while Cp110 levels are mainly regulated via the

ubiquitin-dependent proteasome system during the cell cycle (*D'Angiolella et al., 2010*; *Li et al., 2013*), Cp110 levels in differentiated MCCs are also subject to post-transcriptional repression by microRNAs (miRs) from the *miR-34/449* family (*Song et al., 2014*). Surprisingly, we also found that loss of Cp110 inhibits cilia formation in MCCs (*Song et al., 2014*), suggesting a more complex, and supportive role for Cp110 in ciliogenesis than previously anticipated. A recent report further supports this view, as deletion of *Cp110* exon 5 impairs primary cilia formation in the mouse (*Yadav et al., 2016*).

Here, we use *Xenopus* embryos, whose epidermis provides a readily accessible model to study MCCs of mucociliary epithelia (*Werner and Mitchell, 2012*), as well as other mono-ciliated cells (*Schweickert and Feistel, 2015*). We show that Cp110 localizes to cilia-forming basal bodies and is required for the formation and function of all principal types of cilia (i.e. primary sensory cilia, motile mono-cilia and motile cilia of MCCs). In MCCs, Cp110 is specifically needed for ciliary adhesion complex (*Antoniades et al., 2014*) formation and basal body interactions with the Actin cytoskeleton. Furthermore, we demonstrate that Cp110's opposing roles in ciliogenesis are determined by its multi-domain protein structure. Due to its dual role, optimal Cp110 levels need to be produced to facilitate multi-ciliogenesis. We provide evidence, that optimal regulation of cellular Cp110 levels in MCCs is achieved through a transcriptional/post-transcriptional gene regulatory module, consisting of ciliary transcription factors and miRNAs (*Song et al., 2014*; *Choksi et al., 2014*; *Marcet et al., 2011*; *Chevalier et al., 2015*).

## Results

### Cp110 is required for ciliogenesis at the level of basal body function

To elucidate the effects of *cp110* knockdown on MCC ciliogenesis in detail, we investigated mucociliary clearance and motile cilia function in vivo. Extracellular fluid flow was analyzed by high-speed microscopy and particle tracking of fluorescent beads (*Walentek et al., 2014*). Control embryos generated a directional and robust flow along the epidermis, while Morpholino oligonucleotide (MO)-mediated knockdown of *cp110* caused strongly reduced fluid flow velocities and loss of directionality (*Figure 1A–B*; *Video 1*). Next, we visualized cilia beating directly by injection of *gfp-cfap20* (encoding an axonemal protein) and confocal resonant scanning microscopy (*Turk et al., 2015*). MCCs in control embryos showed directionally uniform and metachronal synchronous ciliary beating, while depletion of Cp110 caused asynchronous beating, reduced motility and randomization of directionality or a complete loss of motility (*Figure 1—figure supplement 1A–B*; *Videos 2–3*). Next, we analyzed basal bodies using the markers Centrin4-RFP (basal body) and Clamp-GFP (ciliary rootlet) (*Park et al., 2008*). In *cp110* morphants, basal bodies aggregated, leading to loss of directional alignment (*Figure 1C*), in turn a prerequisite for directional MCC cilia beating.

To investigate defects in ciliogenesis, we injected *centrin4-cfp* alone or together with *cp110*MO and analyzed cilia formation by immunofluorescence. About 95% of targeted MCCs were fully ciliated in controls, but less than 1% of targeted MCCs were fully ciliated in *cp110* morphants (*Figure 1D*; *Figure 1—figure supplement 2A*). *cp110*MO's effects on MCC ciliation were cell-autonomous as non-targeted MCCs showed normal cilia, suggesting that basal body maturation or function was disrupted, rather than signaling or epithelial morphogenesis. While control MCCs showed basal bodies interspersed into a dense Actin network at the apical membrane, apical Actin formation was disrupted in Cp110-deficient MCCs and a large portion of basal bodies remained deep in the cytoplasm, indicating deficient apical transport of basal bodies (*Figure 1E*; *Figure 1—figure supplement 2B–D*). Loss of basal body transport is predicted to prevent basal body apical docking, alignment, as well as cilia formation (*Marshall, 2008*). In order to gain more insight into the primary versus secondary effects of Cp110 loss, we injected embryos with increasing concentrations of *cp110*MO. These experiments revealed a dose-dependent effect of *cp110* knockdown (*Figure 1—figure supplement 1C–D*). At low doses *cp110*MO caused loss of basal body alignment and mild apical Actin defects, without interfering with basal body apical transport, docking and cilia formation. In contrast, high doses of *cp110*MO primarily interfered with basal body apical transport and prevented cilia formation. Rescue experiments further confirmed the specificity of the MCC phenotype in *cp110* morphants (*Figure 1D–E*; *Figure 1—figure supplement 2*): co-injection of MO-insensitive *cp110* DNA (*cp110-fs*) restored ciliation rates and partially restored apical basal body

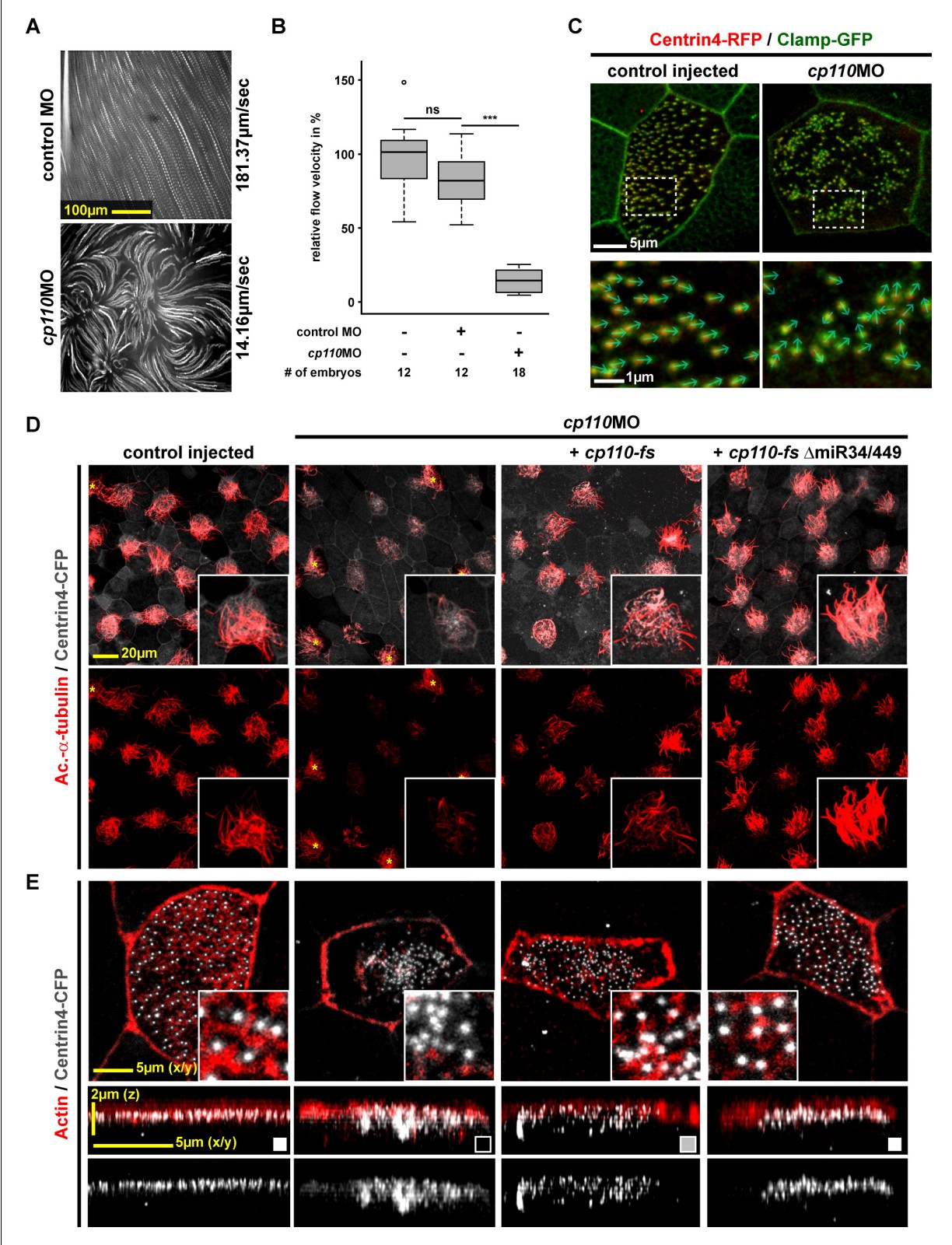

**Figure 1.** Cp110 is required for basal body function in MCC ciliogenesis. (**A**) *cp110* knockdown causes impaired extracellular fluid flow. Control (uninjected controls and control MO injected specimens) and *cp110*MO-injected embryos were analyzed for extracellular fluid flow (10 s projections are shown). (**B**) Velocities were quantified by particle tracking (Related to *Video 1*). ***p<0.001; ns, p>0.05 from Wilcoxon two-sample test. (**C**) Alignment of basal bodies is disrupted in *cp110* morphant MCCs. Centrin4-RFP (basal bodies, red), Clamp-GFP (rootlets, green). Arrows in bottom panels show basal

*Figure 1 continued on next page*

*Figure 1 continued*

body directionality. n embryos/MCCs: control (9/27), *cp110*MO (10/30). (**D**) Knockdown of *cp110* causes severe defects in MCC ciliogenesis which can be rescued by *cp110* DNA co-injection, demonstrated by immunofluorescence for Acetylated-α-tubulin (cilia, Ac.-α-tub., red). Trgeted MCCs were identified by co-injection of *centrin4-cfp*. Non-targeted MCCs (asterisks) produced normal cilia. (Related to *Figure 1—figure supplement 2A*). (**E**) Loss of Cp110 disrupts basal body apical transport and F-actin formation. Basal bodies (Centrin4-CFP, white) and Actin (red) are shown in apical (top row) and lateral (bottom rows) views of individual MCCs. Top views and lateral projections show representative examples (boxes indicate phenotype: white = wt; gray = mild docking defect; black = severe docking defect). (Related to *Figure 1—figure supplement 2B,C*). See also:

The following figure supplements are available for figure 1:

**Figure supplement 1.** Cp110 is required for basal body function in MCC ciliogenesis.

**Figure supplement 2.** Quantification of basal body and ciliogenesis phenotypes in MCCs.

localization. As previously described for Cp110 gain-of-function experiments (*Song et al., 2014*), a more potent *cp110* DNA from which the *miR-34/449* target site was removed (*cp110-fsΔmiR34/449*) showed higher rescue efficiencies.

To confirm the disruptive effect of Cp110 loss-of-function on signaling through primary cilia, we analyzed Hedgehog-dependent gene expression in the developing nervous system (*Dessaud et al., 2008*). *cp110* knockdown reduced expression of both *nkx2.2* and *pax6* confirming impaired Hedgehog signaling in Cp110-depleted *Xenopus* embryos (*Figure 2A*; *Figure 2—figure supplement 1A–C*). We also tested if motile mono-cilia of the *Xenopus* embryonic left-right (LR) organizer (the Gastrocoel Roof-Plate [GRP]) (*Blum et al., 2014*) depend on Cp110 function. GRP ciliation rates were reduced to about 26% in *cp110* morphants, as compared to 85% in control embryos, and the remaining cilia were shorter and more frequently mispolarized (*Figure 2B*; *Figure 2—figure supplement 1D–F*). Ciliary function in the GRP is required for LR-asymmetric gene expression, including *pitx2c,* and loss of Cp110 randomized *pitx2c* gene expression in the lateral plate mesoderm (*Figure 2C*; *Figure 2—figure supplement 1G*).

Taken together, our data revealed the requirement for Cp110 in ciliation of all principal types of cilia during *Xenopus* development and suggested that Cp110 is required at the level of the basal body to promote ciliogenesis.

## Cp110 localizes to cilia-forming basal bodies

We next analyzed Cp110 localization in *Xenopus*, human and mouse MCCs. In all cases, Cp110 localized to cilia-forming basal bodies (*Figure 3A–B,D*; *Figure 3—figure supplement 1C–E*), in addition to its previously described localization to centrosomes. Co-expression of *gfp-cp110* mRNA at levels permitting normal ciliogenesis, together with *centrin4-cfp* (basal body) and *clamp-rfp* (rootlet) further confirmed co-localization of these proteins in apically docked basal bodies *in vivo* (*Figure 3C*). In addition to the predominant GFP-Cp110 localization adjacent to the basal body, smaller amounts were concentrated at the tip of the rootlet (*Figure 3C'*). These novel localization patterns of overexpressed GFP-Cp110 at basal bodies were confirmed by analysis of endogenous Cp110 in MCCs of *in vitro* cultured human airway epithelial cells (HAECs): Cp110 was found interspersed into the apical Actin network and found at the base of MCC cilia (*Figure 3B,D*). Endogenous Cp110 also co-localized with and extended Centrin1 in human MCCs (*Figure 3E*), and super-resolution structured illumination microscopy (3D-SIM) verified Cp110 localization adjacent to the

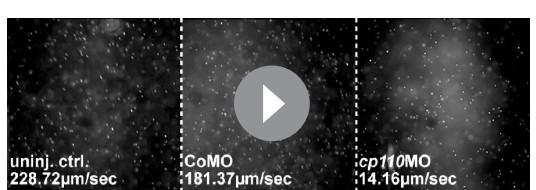

**Video 1.** Cp110 is required for extracellular fluid flow in the *Xenopus* mucociliary epidermis. Extracellular fluid flow over the *Xenopus* embryonic epidermis was analyzed at stage 32 by time-lapse imaging of fluorescent beads. Knockdown of *cp110* caused severely reduced fluid flow velocity (*cp110*MO; 14.16 μm/s) and loss of directionality, as compared to control MO-injected (CoMO; 181.37 μm/s) and uninjected (uninj. ctrl.; 228.72 μm/s) specimens. Movie plays at 1x speed. Related to *Figure 1A*.

**Video 2.** Cp110 is required for metachronal synchronous ciliary beating in MCCs. Embryos were injected with *cfap20-gfp* to visualize ciliary axonemes of epidermal MCCs at stage 32 by resonant confocal microscopy. Anoptical section along the MCC apical-basal axis is shown (apical up). Control MCCs (uninj. ctrl.) showed a metachronal synchronous beating pattern of cilia. Knockdown of *cp110* (*cp110*MO) disrupted the metachronal synchronous beating pattern and caused reduced motility in MCC cilia. Movie plays at 1x speed. Related to *Figure 1—figure supplement 1A–B*.

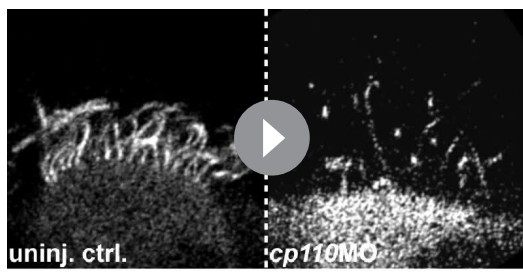

**Video 3.** Cp110 is required for unidirectional ciliary beating and ciliary motility in MCCs. Embryos were injected with *cfap20-gfp* to visualize ciliary axonemes of epidermal MCCs at stage 32 by resonant confocal microscopy. Horizontal optical section through the MCC ciliary tuft is shown. Control MCCs (uninj. ctrl.) showed a unidirectional beating pattern of cilia. Knockdown of *cp110* (*cp110*MO) caused loss of directionality and reduced motility in MCC cilia. Movie plays at 1x speed. Related to *Figure 1—figure supplement 1A–B*.

basal body (*Figure 3F*). In conclusion, we present new localization sites of Cp110 at basal bodies, which are distinct from its previously described location at the distal end of centrioles, where Cp110 inhibits axoneme elongation.

Expression of *gfp-cp110* together with *centrin-cfp* in the GRP also confirmed localization of Cp110 to cilia-forming basal bodies in motile mono-cilia (*Figure 3—figure supplement 2A*). GRP cells were more sensitive to *gfp-cp110* overexpression, possibly because of the limited number of centrioles/basal bodies as compared to MCCs. This led to another interesting observation: GFP-Cp110 basal body levels inversely correlated with GRP cilia length, i.e. high expression levels inhibited cilia, intermediate levels caused shorter cilia, and low levels permitted normal cilia formation (*Figure 3—figure supplement 2B*). This implies that, in contrast to previous reports, Cp110 might play a role in ciliary length control. Our hypothesis was further supported by the observation that Cp110 can localize to ciliary tips in some GRP cells (*Figure 3—figure supplement 2C*). This observation might be related to the requirement to coordinately resorb cilia from GRP cells after the LR-body axis is specified.

Interestingly, overexpression of *gfp-cep97*, another negative regulator of ciliogenesis and Cp110-interacting partner (*Spektor et al., 2007*), revealed specific localization of GFP-Cep97 to centrosomes of epidermal cells, however no localization to cilia-forming basal bodies in MCCs was observed (*Figure 3—figure supplement 3A–C*). Furthermore, GFP-Cep97 was not able to suppress cilia formation in MCCs (*Figure 3—figure supplement 3D*). This data further supported a specific role for Cp110 in ciliogenesis, independent of Cep97.

In summary, our data demonstrate three novel locations of Cp110 accumulation in basal bodies and cilia, in addition to its previously described localization to distal ends of centrioles (*Figure 3—figure supplement 4*): (1) Adjacent to cilia-forming basal bodies, (2) at rootlets, and (3) at the tip of cilia.

## Cp110 is required for ciliary adhesion complex formation

In addition to Cp110 localization to MCC basal bodies and rootlets, we also observed Cp110 localization to basal bodies during stages of apical basal body transport (preceding ciliogenesis) (*Figure 3—figure supplement 1A*) as well as an asymmetry in basal body Cp110 levels along the anterior-posterior axis in some MCCs (early MCC stages) (*Figure 3—figure supplement 1B*). The same types of localization patterns were reported for ciliary adhesion complex components, Focal Adhesion Kinase (FAK), Vinculin and Paxillin (*Antoniades et al., 2014*). In MCCs, these are required for basal body binding to F-actin. Furthermore, loss of Cp110 phenocopied loss of FAK in *Xenopus*

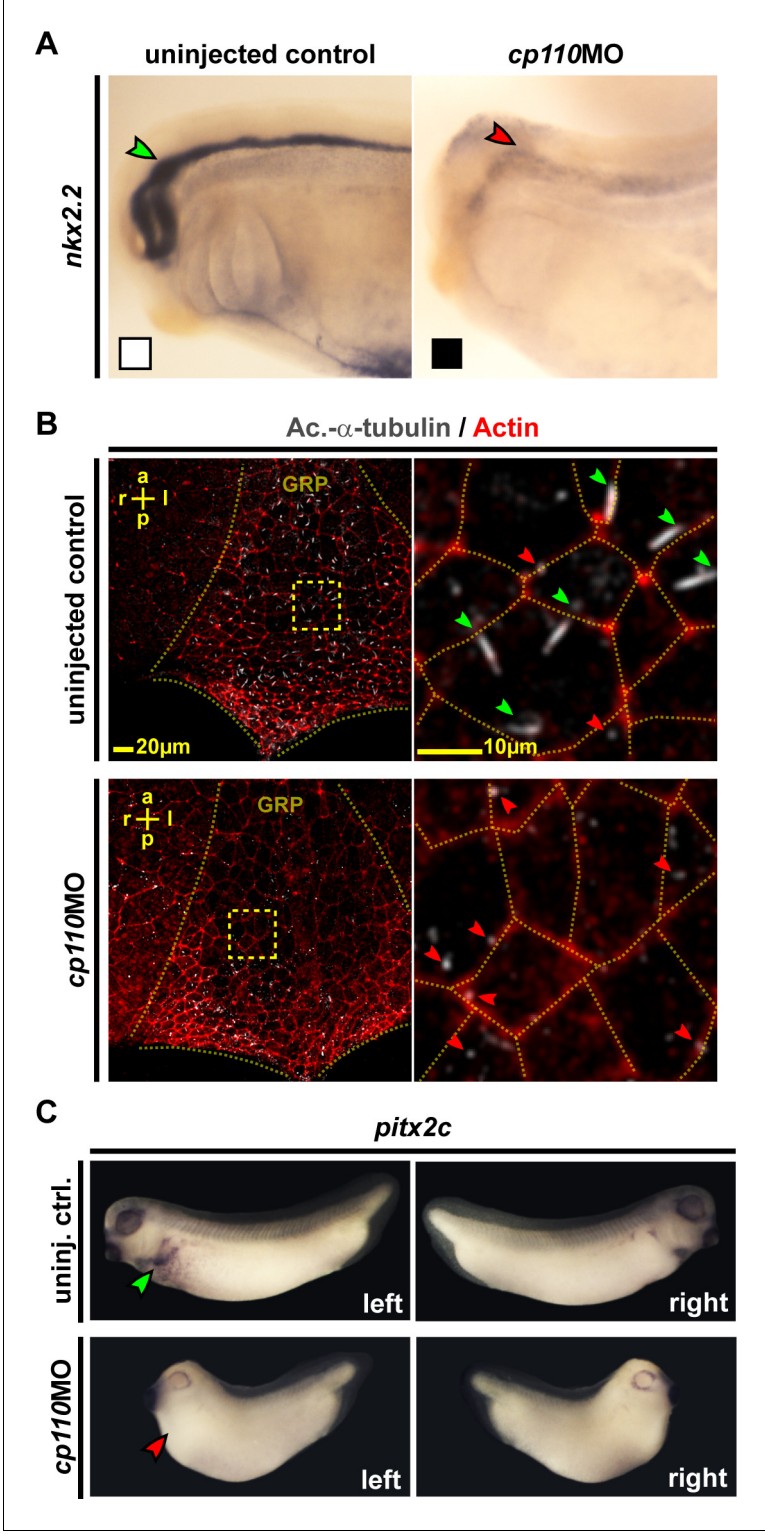

**Figure 2.** Cp110 is required for primary and motile monocilia. (**A**) Cp110 is required for Hedgehog signaling-dependent *nkx2.2* expression. Whole-mount in situ hybridization (WMISH) staining for *nkx2.2* expression in the neural tube. Normal expression indicated by green arrowhead, reduced expression indicated by red arrowhead. Related to *Figure 2—figure supplement 1A–C* (white and black boxes indicate normal and reduced expression, respectively in graph *Figure 2—figure supplement 1A*). (**B**) Cp110 is required for GRP cilia. Immunofluorescent staining for cilia (Ac.-α-tub., white) and cell borders (Actin, red). Green arrowheads, normal cilia; red arrowheads, defective cilia. Related to *Figure 2—figure supplement 1D–F*. (**C**) *cp110*MO interferes with left side specific
*Figure 2 continued on next page*

*Figure 2 continued*

*pitx2c* expression in the lateral plate mesoderm as shown by WMISH. Green arrowhead, normal/left expression; red arrowhead, absent expression. Related to *Figure 2—figure supplement 1G*. See also:
The following figure supplement is available for figure 2:

**Figure supplement 1.** Cp110 is required for primary and motile monocilia.

MCCs. We therefore explored whether Cp110 might be required for ciliary adhesion complex formation or function.

First, we analyzed localization of Cp110 and FAK in *Xenopus* MCCs. Both Cp110 and FAK localized to posterior sites at the basal body and the rootlet, with FAK extending the Cp110 basal body domain (*Figure 4 A*; *Figure 4—figure supplement 1A*). Co-immunoprecipitation (co-IP) experiments using overexpressed FLAG-Cp110 in combination with FAK-GFP or Centrin-GFP further suggested an interaction between Cp110, FAK and Centrin4 (*Figure 4 B*; *Figure 4—figure supplement 2E*). In contrast to the overexpression tests in *Xenopus*, we were not able to convincingly co-IP endogenous FAK using two commercially available anti-Cp110 antibodies in ciliated HAECs (not shown). Therefore, although our data suggest that Cp110 and ciliary adhesion components localize to the same sites at basal bodies, this co-localization could rely on additional intermediate protein complexes, similar to the situation described for Cp110 interactions with Centrin at centrioles (*Tsang et al., 2006*).

To test the hypothesis that Cp110 is functionally required for ciliary adhesion complex formation or function, we depleted Cp110 in MCCs and analyzed FAK localization to basal bodies. In control MCCs, FAK-GFP normally accumulated at basal bodies and rootlets, and this localization was strongly reduced in Cp110-deficient MCCs (*Figure 4C*). Higher magnification revealed residual FAK-GFP levels at basal bodies in *cp110* morphants (*Figure 4D*), which was further confirmed by analyzing the ratio of fluorescent intensity of FAK-GFP/Centrin4-CFP: on average, FAK-GFP levels were reduced to about 30% in Cp110-deficient MCCs as compared to controls (*Figure 4F*). We also addressed whether other ciliary adhesion components were affected by Cp110 depletion. As with FAK-GFP, both Vinculin-GFP and Paxillin-GFP levels were strongly reduced in *cp110* morphant basal bodies and rootlets, and Vinculin-GFP was also reduced at the apical membrane (*Figure 4—figure supplement 2A–Ds*). Ciliary adhesion components appeared to be more dramatically reduced at rootlets than at basal bodies. To confirm that loss of ciliary adhesions was not primarily caused by a loss of the rootlet, we analyzed FAK-GFP localization in *cp110* morphants which were triple-injected with *centrin4-cfp*, the rootlet marker *clamp-rfp* and *FAK-gfp*. These experiments confirmed strong reduction of FAK-GFP from basal bodies and rootlets, while Centrin4-CFP and Clamp-RFP were still present, at least in basal bodies residing close to the apical membrane (*Figure 4—figure supplement 1B*). These results are further supported by published mouse data, which demonstrated that Rootletin localizes to Cp110-deficient basal bodies (*Yadav et al., 2016*), and that Rootlet-deficient basal bodies still form cilia in mono- and multi-ciliated cells (*Yang et al., 2005*).

Lastly, we also tested if Cp110 was not only required for ciliary adhesion complex formation, but if *cp110* overexpression could be sufficient to enhance recruitment of FAK-GFP to basal bodies and rootlets. Indeed, our results indicate that exogenous Cp110 can recruit additional FAK-GFP in a dose-dependent manner (*Figure 4E–F*).

We conclude from these experiments that Cp110 is required for the normal formation of ciliary adhesion complexes in MCCs, and suggest that loss of FAK from basal bodies and rootlets causes the observed defects in basal body apical transport, docking and alignment, as well as the loss of apical Actin network formation (*Figure 4—figure supplement 3*).

## Distinct protein domains promote centriolar versus ciliary functions of Cp110

Cp110 is a multi-domain protein regulating multiple processes during the cell cycle via interaction with distinct partners (*Tsang and Dynlacht, 2013*). Indeed, the two opposing roles of Cp110 in cilia formation might be promoted by specific protein domains, interacting with different protein

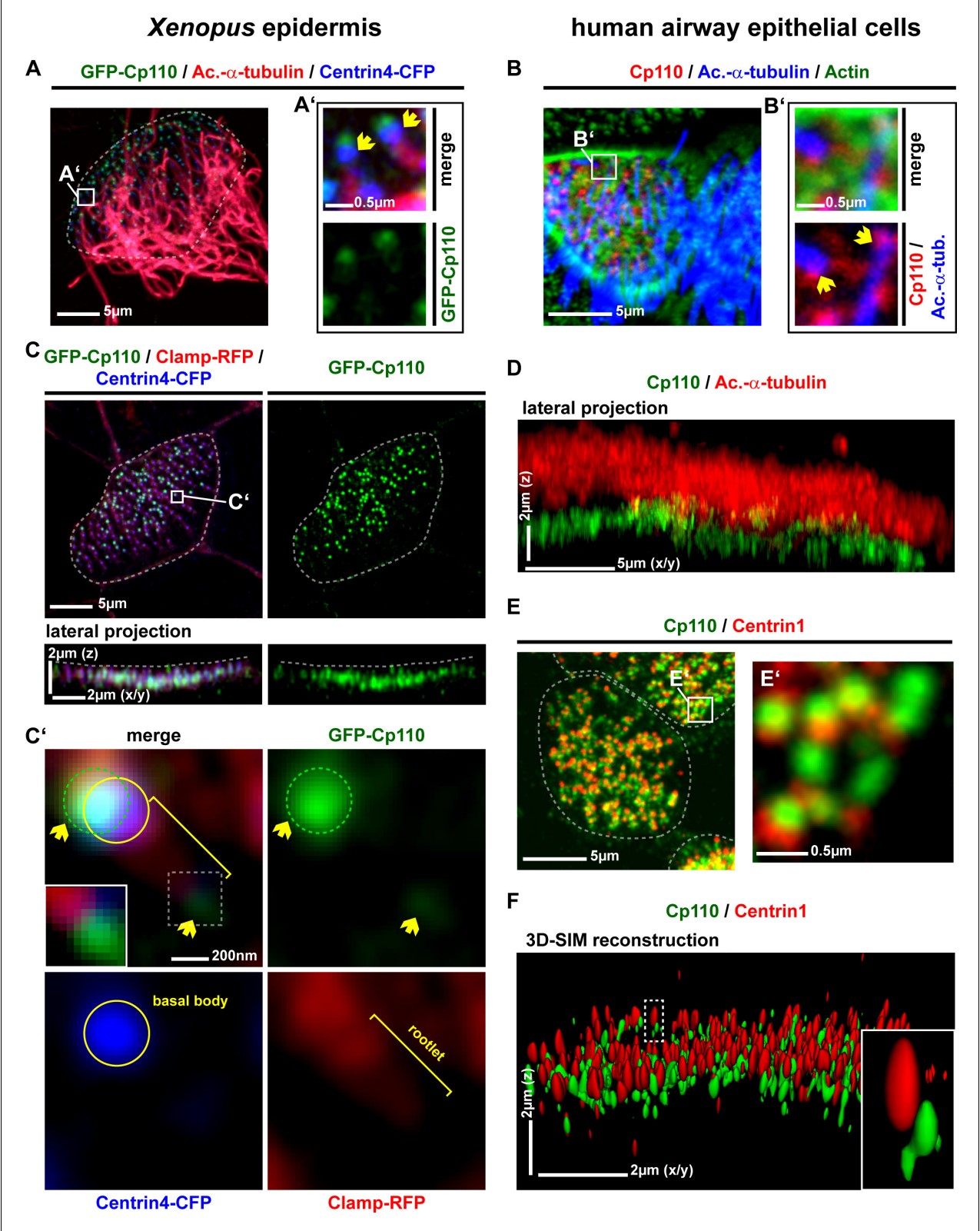

**Figure 3.** Cp110 localizes to cilia-forming basal bodies in MCCs. (**A–D**) Cp110 localizes to cilia-forming basal bodies in *Xenopus* epidermal (**A**, **C**) and human airway epithelial cell (HAEC) (**B**, **D**) MCCs. (**A**) *gfp-cp110* (green) was expressed at levels permitting ciliogenesis, together with *centrin4-cfp* (basal bodies, blue). Immunofluorescent staining (Ac.-α-tub., red) was used to visualize cilia. (**B**) Immunofluorescent staining for endogenous Cp110 (red), cilia (Ac.-α-tub.; blue) and Actin (green) in MCCs (n donors = 1, n MCCs = 4). Yellow arrows in A' and B' indicate the base of cilia. (**C**) Apical view (top) of

*Figure 3 continued on next page*

Figure 3 continued

individual MCC co-injected with *gfp-cp110* (green), *centrin4-cfp* (blue) and *clamp-rfp* (red) to visualize Cp110, basal bodies and rootlets, respectively. Localization of basal bodies to the apical membrane is shown in lateral projection (bottom). n embryos/MCCs: (4/18). (C') High-magnification analysis of GFP-Cp110 (green, indicated by yellow arrows and green circle) binding to an individual basal body from the MCC shown in (C) (basal body and rootlet are indicated). Inset shows rootlet domain (dashed box) with increased brightness. (D) Lateral projection of MCC stained for endogenous Cp110 (green) and cilia (Ac.-α-tub.; red). n donors = 1, n MCCs = 12 (same samples as in *Figure 3—figure supplement 1C*) (E–F) Endogenous Cp110 (green) and Centrin 1 (red) staining shows Cp110 adjacent to MCC basal bodies by confocal microscopy (E) and 3D-SIM imaging (F). n donors = 1, n MCCs = 3 each for confocal and 3D-SIM.

The following figure supplements are available for figure 3:

**Figure supplement 1.** Cp110 localizes to cilia-forming basal bodies in MCCs.

**Figure supplement 2.** Cp110 localizes to cilia-forming basal bodies and ciliary tips of monociliated GRP cells.

**Figure supplement 3.** Cep97 does not localize to cilia-forming basal bodies.

**Figure supplement 4.** Schematic depiction of Cp110 localization sites at centrioles, basal bodies and cilia.

complexes. We first analyzed the *Xenopus tropicalis* (Xt) Cp110 sequence and found that the deposited reference sequence (matching BC167469) predicted a shorter product than in other species. Compared to the Xt7.1 genome sequences (*Hellsten et al., 2010*; *Karpinka et al., 2014*), the clone has a frameshift producing a premature stop codon and a truncated Cp110 (Cp110-FS) (*Figure 5D*). Restoration of the missing Adenine generated a full-length Cp110 version of 962 amino acids, which contains the domains described for human Cp110 (*Chen et al., 2002*) at similar positions, including two coiled-coil domains (CCDs), CDK phosphorylation sites, Cyclin binding domains (RXL) and CaM-binding domains (*Figure 5D*). Interestingly, the KEN box motif, which is required for proteasomal targeting of Cp110, is positioned more towards the N-terminus in Xt Cp110 than in human Cp110, which instead contains an additional destruction-box (D box) with the same function at a similar position. Judging from our functional experiments employing *cp110-fs* and *cp110-fsΔmiR34/449* and correction of the sequence, we conclude that (a) Cp110-FS is still largely functional, (b) the *miR-34/449* target site is located within the wild-type coding sequence, and (c) the domain structure of Cp110 is highly conserved among vertebrates.

Next, we generated GFP-tagged *cp110* deletion constructs and tested their ability to inhibit cilia formation in MCCs. Most deletion constructs inhibited cilia formation at similar rates to full-length GFP-Cp110, but not a construct missing both CCDs (GFP-Cp110ΔCCD1&2), which had very mild effects on cilia (*Figure 5A*; *Figure 5—figure supplement 1A–B*). CCDs can facilitate intramolecular interactions as well as intermolecular interactions during complex formation (*Kuhn et al., 2014*; *Burkhard et al., 2001*; *Salisbury, 2003*). We therefore deleted Cp110 CCDs separately and tested these constructs for cilia inhibition. Deletion of the first CCD alone (GFP-Cp110ΔCCD1) did not affect cilia inhibition, while deletion of the second CCD (GFP-Cp110ΔCCD2) only weakly affected cilia suppression (*Figure 5—figure supplement 1B*). This suggested that CCDs overlap in activity. For all constructs, effects were cell-autonomous, and non-targeted MCCs formed cilia comparable to uninjected controls. We conclude from this data, that (a) expression of all deletion constructs results in the production of functional protein, which we have confirmed by immunoblotting for GFP-Cp110 constructs at relevant stages (*Figure 5—figure supplement 1C*), (b) the CCDs of Cp110 inhibit ciliation in a redundant manner, and (c) Cp110 CCDs likely mediate binding to protein complexes at the basal body which support cilia inhibition.

In addition to effects on ciliogenesis, we observed effects in non-MCC epidermal cells upon over-expression of *gfp-cp110* constructs (*Figure 5A*; *Figure 5—figure supplement 1A* and *2A–D*): Most constructs frequently induced multiple GFP-Cp110 foci per cell and enlargement of cells, while both effects were absent upon *gfp-cp110ΔCCD1&2* overexpression. GFP-Cp110 constructs also strongly localized to centrioles, but centriolar localization of GFP-Cp110ΔCCD1&2 was relatively weak. Formation of supernumerary centrioles and resulting defects in cytokinesis and chromosome separation were previously described upon Cp110 overexpression in cycling cells (*Tsang et al., 2006*;

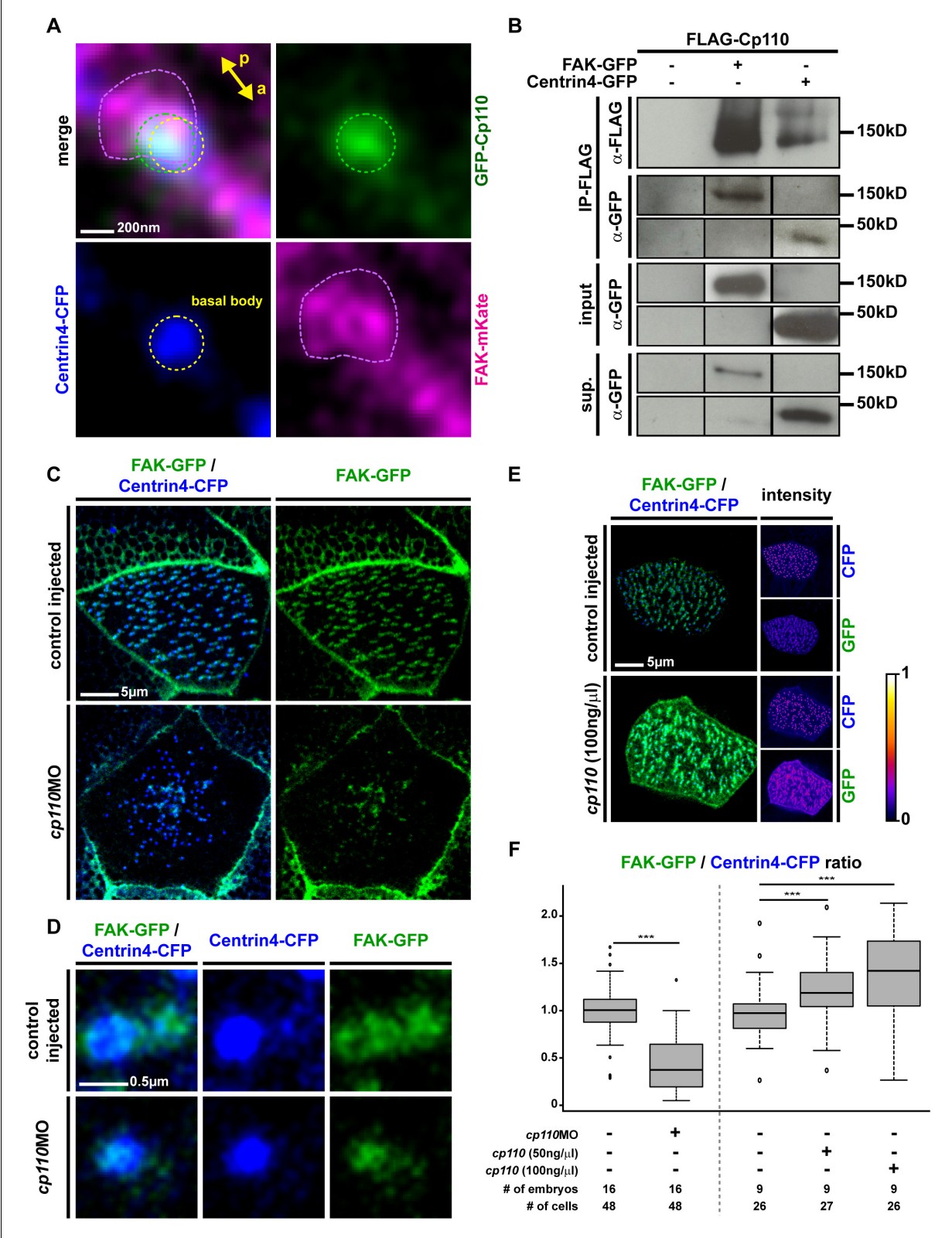

**Figure 4.** Cp110 is required for ciliary adhesion complex formation in MCCs. (**A**) Expression of *gfp-cp110* (green) at concentrations permitting ciliogenesis, *FAK-mKate* (magenta), and *centrin4-cfp* (blue) revealed polarized posterior localization of GFP-Cp110 and FAK-mKate adjacent to the basal body. Note that FAK-mKate overlaps with GFP-Cp110, but extends past GFP-Cp110 in the posterior direction. n embryos/MCCs (7/28). Related to **Figure 4—figure supplement 1A**. (**B**) Western blot analysis of co-immunoprecipitation (co-IP) using Flag-Cp110. FLAG-Cp110 (~140kD) detected by
*Figure 4 continued on next page*

*Figure 4 continued*

anti-FLAG antibody (α-FLAG). FAK-GFP (~150kD) and Centrin4-GFP (~45kD) detected by anti-GFP antibody (α-GFP). Co-IP, IP-FLAG; input samples, input; supernatant samples, sup. (n = 2). Related to *Figure 4—figure supplement 2E*. (C–D) Cp110 is required for FAK binding to MCC basal bodies. (C) Mix of *FAK-gfp* (green) and *centrin4-cfp* (blue) mRNAs was injected (± *cp110*MO). Quantification shown in (F). n embryos/MCCs: control (16/48), *cp110*MO (16/48). (D) Magnification of individual basal body from C. (E) Overexpression of Cp110 caused increased localization of FAK-GFP to basal bodies (Centrin4-CFP, blue). Heatmaps of CFP and GFP intensity levels shown next to merged immunofluorescent images. Color code shown right. Quantification shown in (F). n embryos/MCCs: control (9/26), 50 ng/µl (9/27), 100 ng/µl (9/26). (F) Quantification of FAK-GFP to Centrin4-CFP ratios in controls, *cp110* morphants and after overexpression of *cp110* (at 50 ng/µl and 100 ng/µl concentrations). ***p<0.001 from Wilcoxon two-sample test.

The following figure supplements are available for figure 4:

**Figure supplement 1.** Cp110 is required for ciliary adhesion complex formation in MCCs.

**Figure supplement 2.** Cp110 is required for ciliary adhesion complex formation in MCCs.

**Figure supplement 3.** Schematic representation of summary model of the roles of Cp110 in MCC ciliation.

*Chen et al., 2002*); therefore our findings suggest that Cp110 CCDs are required for centriolar functions, including suppression of ciliogenesis.

Next, we overexpressed *gfp-cp110* deletion constructs with *centrin-cfp* and *clamp-rfp* to investigate MCC basal body behavior. Basal bodies in control MCCs were uniformly aligned and spaced appropriately, while overexpression of full-length *gfp-cp110* caused mild alignment defects as well as aggregation of basal bodies (*Figure 5B*). The same defects were observed with most deletion constructs, but not in *gfp-cp110ΔCCD1&2* overexpressing MCCs (*Figure 5 B*; *Figure 5—figure supplement 1D*). Interestingly, the negative effects on basal bodies and cell size were variable among cilia-inhibiting constructs (*Figure 5—figure supplement 2E*): Most prominently, deletion of a central domain (GFP-Cp110ΔCentral) containing most phosphorylation sites, RXL and KEN domains (proteasome targeting motifs), induced much stronger effects than full-length Cp110 and protein levels were elevated in comparison to other constructs (*Figure 5A–B*; *Figure 5—figure supplement 1C*). Collectively, the data suggested that the central domain deletion generated a hypermorphic protein, which was released from negative regulation by the proteasomal machinery (*Tsang and Dynlacht, 2013*). Overexpression of *gfp-cp110ΔCentral* also caused the formation of extremely enlarged MCCs and other epithelial cells, which were frequently polynucleated, indicating severe cytokinesis defects (*Figure 5—figure supplement 2A*). Abnormal nuclei were associated with multiple GFP-Cp110 foci, which also contained Centrin4-CFP, indicating presence of supernumerary centrioles (*Figure 5—figure supplement 2B*). In MCCs, *gfp-cp110ΔCentral* induced strong aggregation and an increased number of basal bodies (*Figure 5—figure supplement 2C–D*), likely due to the presence of supernumerary centrioles at the onset of deuterosome-mediated centriole amplification.

Finally, we investigated GFP-Cp110 localization to basal bodies and rootlets. All constructs were able to localize to basal bodies, although we detected differences in relative binding to different parts, especially when comparing full-length GFP-Cp110 to GFP-Cp110ΔCentral and GFP-Cp110ΔCCD1&2 (*Figure 5C*; *Figure 5—figure supplement 1E*). GFP-Cp110 overlapped mainly with the distal and posterior basal body and localized at much lower levels to the tip of the rootlet, while GFP-Cp110ΔCentral preferentially localized to the basal body with reduced levels at the rootlet. Conversely, GFP-Cp110ΔCCD1&2 localization was stronger at the rootlet as compared to the basal body. Interestingly, overexpression of each construct negatively affected apical Actin formation, while the apical transport of basal bodies occurred largely normally unless aggregation of basal bodies was observed (*Figure 5—figure supplement 1F*).

In summary, these results support the conclusion that at centrioles and basal bodies Cp110 CCDs promote binding to centriolar-type protein complexes to mediate cilia inhibition, while other domains allow Cp110 interaction with cilia-promoting or cell-cycle regulatory complexes (*Figure 5—figure supplement 3*).

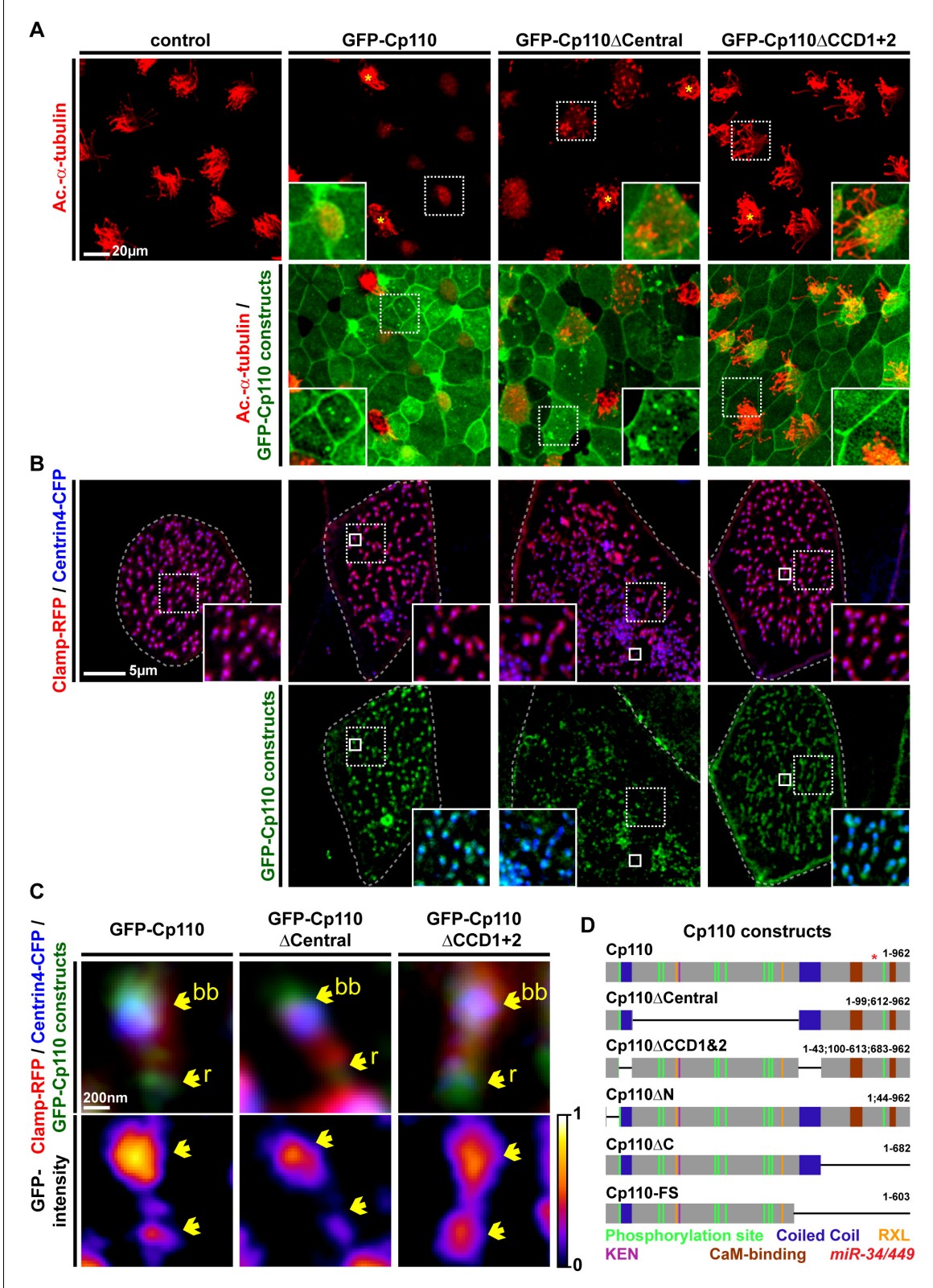

**Figure 5.** Cp110 coiled-coil domains are required for cilia inhibition and centriolar functions. (**A**) Cp110 coiled-coil domains are required for MCC cilia inhibition and formation of supernumerary centrioles. Controls and embryos injected with full-length *gfp-cp110, gfp-cp110ΔCentral* or *gfp-cp110ΔCCD1+2* (all green) were analyzed for ciliation by immunofluorescent staining (Ac.-α-tub., red). Upper panels: red fluorescence channel; merge channel insets show magnifications outlined by dashed boxes. Lower panels: green/red merge; insets show magnifications of non-MCCs outlined by dashed boxes. *Figure 5 continued on next page*

*Figure 5 continued*

Related to *Figure 5—figure supplement 1A–B*. (**B**) Cp110 coiled-coil domains are required for basal body aggregation. Embryos injected with *centrin4-cfp* (basal bodies, blue), *clamp-rfp* (rootlets, red), and either full-length *gfp-cp110* or *gfp-cp110ΔCentral* or *gfp-cp110ΔCCD1+2* (green). Upper panels: red/blue merge; insets show magnifications of basal bodies outlined by dashed boxes. Lower panels: green/blue merge; insets show magnifications of basal bodies outlined by dashed boxes. n embryos/MCCs: control (7/21), *gfp-cp110* (7/21), *gfp-cp110ΔCentral* (7/21), *gfp-cp110ΔCCD1+2* (7/21). Related to *Figure 5—figure supplement 1D*. (**C**) GFP-Cp110 constructs show different localization patterns at basal bodies (bb) and rootlets (r). Upper panels: Individual basal bodies from MCCs shown in (**B**) (solid boxes). Lower panels: heat maps of GFP-Cp110 intensity. Color code shown right. Related to *Figure 5—figure supplement 1D*. (**D**) Cp110 constructs generated in this study. Different colors indicate predicted functional domains. Green, CDK phosphorylation sites; blue, coiled-coil domains; yellow, Cyclin binding domain (RXL); pink, KEN domain (proteasomal degradation); brown, CaM-binding domains; red asterisk indicates the position of *miR-34/449* target site in the *cp110* mRNA. See also:

The following figure supplements are available for figure 5:

**Figure supplement 1.** Cp110 coiled-coil domains are required for cilia inhibition and centriolar functions.

**Figure supplement 2.** Cp110 central domain deletion enhances centriolar and basal body phenotypes.

**Figure supplement 3.** Schematic representation of Cp110 domains and their proposed function.

## Optimal Cp110 levels are achieved by a transcriptional/post-transcriptional regulatory module in MCCs

Since the dual function of Cp110 in promoting and limiting ciliogenesis is determined by its protein structure and concentration, Cp110 levels need to be tightly controlled to generate optimal cellular quantities for cilia formation. In MCCs, a conserved transcriptional cascade regulates ciliation (*Choksi et al., 2014*). Notch signaling inhibition activates *multicilin (mci*; or *MCIDAS* in humans) (*Stubbs et al., 2012*). Mci forms a ternary complex with E2F-4 or -5 and Dp1 to activate downstream ciliary transcription factors, including *rfx2, foxj1* and *myb* (*Chung et al., 2014*; *Ma et al., 2014*). Together with RFX2 and Foxj1, the Mci complex regulates expression of core multi-ciliogenesis genes. Because Cp110 is indispensable for ciliogenesis, we tested whether its expression is regulated through the MCC transcriptional program. RNA-sequencing (RNA-seq) was performed on manipulated animal cap explants that develop into mucociliary organoids in culture (*Werner and Mitchell, 2012*), and successful manipulation was monitored by assessing expression levels of *foxj1*. During MCC specification stage (st. 16), inhibition of Notch signaling (*su(h)-dbm*) or stimulation of multi-ciliogenesis (*mci*) resulted in strongly increased *cp110* expression as compared to overactivation of notch signaling (*notch-icd*) or inhibition of multi-ciliogenesis (*dominant-negative- (dn-)mci*) (*Figure 6A*; *Figure 6—figure supplement 1A*). Furthermore, ciliary transcription factors bind to the *cp110* locus; Chromatin Immunoprecipitation and DNA-sequencing (ChIP-seq) showed binding of E2F4, RFX2 and Foxj1 to the transcriptional start site of *cp110*, and additional Foxj1 binding to intronic regions of *cp110* (*Figure 6B*). Therefore, Cp110 is a core multi-ciliogenesis protein, which is regulated by ciliary transcription factors in MCCs.

While manipulation of the MCC cascade showed large changes in *cp110* transcript levels at the MCC specification stage (st. 16), these did not persist. At the ciliogenesis stage (st. 25), quantitative RT-PCR (qPCR) showed that *cp110* returned to normal levels, unless *miR-34/449s* were knocked down simultaneously (*Figure 6C*). This suggested that *miR-34/449* expression might also be activated by ciliary transcription factors. We therefore analyzed expression of *miR-34/449* after manipulation of the MCC cascade at stage 25 by qPCR (*Figure 6D*). Like *foxj1* expression, *miR-34b/c* and *miR-449 a/b/c* expression was up- or down-regulated by inhibition of Notch signaling or *dn-mci* injection, respectively. In contrast, expression of *miR-34a* from a third genomic locus was not affected. In agreement with our qPCR results, ChIP-seq and RNA-seq at stage 16 revealed ciliary transcription factor binding and changes in expression for *miR-34b/c* as well as *miR-449a/b/c* (expressed from *cdc20b* intron 2), but not for *miR-34a* (*Figure 6 E–F*; *Figure 6—figure supplement 1B*)

Taken together, we conclude that *cp110* and five of the six *miR-34/449s* are co-activated by ciliary transcription factors during MCC specification stages, which confers their robust expression. Expression of *miR-34/449s* in MCCs then represses *cp110* at the post-transcriptional level preventing

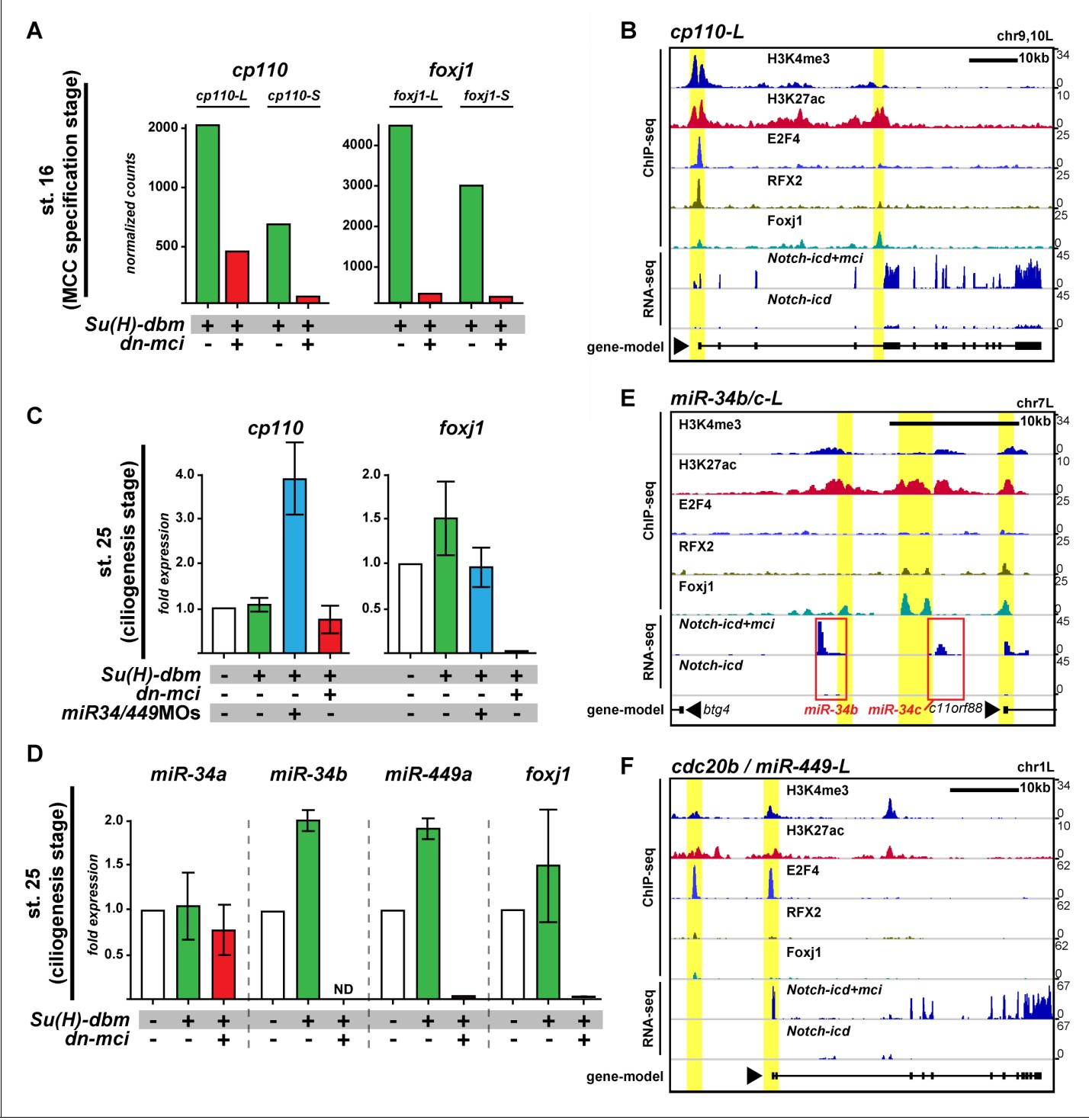

**Figure 6.** Cp110 levels in MCCs are controlled by ciliary transcription factors and *miR-34/449* microRNAs. (A) *cp110* expression in MCCs is regulated through the MCC signaling/transcriptional cascade. Embryos were injected with *Su(H)-dbm* to stimulate MCC induction (green) or with *Su(H)-dbm* and *dominant-negative multicilin (dn-mci)* to prevent MCC induction (red). RNA-sequencing (RNA-seq) was performed at MCC specification stage (st. 16). Normalized counts are shown as bar graphs. n = 2. Related to *Figure 6—figure supplement 1A*. (B) *cp110* expression is activated by ciliary transcription factors. Chromatin immunoprecipitation and DNA-sequencing (ChIP-seq; upper five lanes) and RNA-seq (bottom two lanes) at stage 16. Embryos were injected with *Notch-icd* to inhibit MCC induction or together with *multicilin (mci)* to induce MCCs. ChIP-seq using antibodies to mark active chromatin (Histone H3 lysine tri-methylation, H3K4me3; Histone H3 lysine acetylation, H3K27ac), E2F4 binding (E2F4), RFX2 binding (RFX2), and Foxj1 binding (Foxj1) are shown. A gene model is shown in bottom lane. ChIP-seq peaks are indicated by a yellow background. (C) *cp110* levels at

*Figure 6 continued on next page*

*Figure 6 continued*

ciliogenesis stage (st. 25) are controlled by *miR-34/449* miRNAs. For quantitative RT-PCR analysis (qPCR), manipulations were performed as described in (**A**) (green and red bars). Additionally, *miR-34/449s* were knocked down (*miR-34/449*MO, blue bar). The uninjected control was set to 1. n = 2. (**D**) *miR-34/449* family members are regulated through the conserved MCC signaling/transcriptional cascade. qPCR analysis for *miR-34/449* expression was performed as described (**C**). ND, not detected. n = 2. (**E–F**) Expression of miRNAs *miR-34b/c* and *miR-449a-c* is activated by ciliary transcription factors. ChIP-seq and RNA-seq was performed as described in (**B**). miRNA location in (**E**) is indicated by red box. *miR-449a-c* are expressed from *cdc20b* intron 2. Related to ***Figure 6—figure supplement 1B***. The *foxj1* expression analysis confirmed successful manipulation in (**A**, **C**) and (**D**). Error bars represent s.e.m. in (**C**) and (**D**).

The following figure supplements are available for figure 6:

**Figure supplement 1.** Cp110 levels in MCCs are controlled by ciliary transcription factors and *miR-34/449* microRNAs.

**Figure supplement 2.** Model of the transcriptional/post-transcriptional regulatory module required to achieve optimal Cp110 levels in MCC ciliogenesis.

excess buildup of Cp110 at ciliogenesis stages. Therefore, optimal Cp110 levels in MCC ciliogenesis are generated by a gene regulatory module consisting of ciliary transcription factors and MCC-specific miRNAs from the *miR-34/449* family (***Figure 6—figure supplement 2***).

## Discussion

Here we show that Cp110 localizes posteriorly to cilia-forming basal bodies as well as to rootlets in MCCs, in addition to its well-described localization to distal ends of centrioles. A similar low-level localization of endogenous Cp110 can be also observed in primary cilia of RPE-1 cells (see also Figure 4C in *Tanos et al. 2013*). Furthermore, we demonstrate that optimal cellular levels of Cp110 are required for cilia formation. The general conclusion that Cp110 is required for ciliogenesis is in line with our previous report (*Song et al., 2014*) as well as a recent study in mice, in which *Cp110* was knocked out (*Yadav et al., 2016*).

Cp110's role in cilia suppression was proposed to be mediated by distal end capping of the basal body/centriole (*Spektor et al., 2007*; *Kobayashi et al., 2011*). In this work, we provide evidence that, additionally, loss of Cp110 prevents formation of ciliary adhesion complexes, which in turn mediate interactions of basal bodies and rootlets with F-actin in MCCs and thereby promote cilia formation. As previously described for knockdown of *FAK* (*Antoniades et al., 2014*), knockdown of *cp110* exerts dose-dependent effects on basal bodies and cilia in MCCs: At low concentrations, these effects include loss of sub-apical Actin-dependent basal body alignment and mild defects in apical Actin formation, but nonetheless successful ciliogenesis. At high concentrations, basal bodies fail to migrate to the apical membrane, which prevents apical docking, apical Actin formation and ciliogenesis. Conversely, overexpression of *cp110* leads to increased basal body/rootlet levels of FAK. This indicates that Cp110 is required for ciliary adhesion complex recruitment. It remains to be seen if Cp110 is able to directly interact with FAK or if it requires additional intermediate protein complexes, as seems to be the case for Cp110 interactions with Centrins.

Importantly, neither Cp110 localization to basal bodies nor interactions of basal bodies with F-actin are strict requirements for cilia formation. Disruption of F-actin in quail oviduct MCCs prevents apical transport of basal bodies, but eventually basal bodies dock to cytoplasmic membranes and form aberrant intracellular cilia (*Boisvieux-Ulrich et al., 1990*). Furthermore, ciliary vesicles (CVs) seem to localize apically even in the absence of basal body docking in MCCs (*Park et al., 2008*). Therefore, we propose that apical basal body transport promotes efficient basal body fusion with CVs by facilitating spatial proximity. This interpretation is supported by the finding that about 10% of embryonic fibroblasts were capable of CV fusion and cilia formation in *Cp110* knockout mice, which should lack Cp110 altogether (*Yadav et al., 2016*). Alternatively, Cp110 might promote basal body fusion with CVs through independent interactions.

We further propose that Cp110 could contribute to ciliary length control and coordinated cilia resorption. On the one hand, increased Cp110 levels at the base of cilia are correlated with shorter GRP cilia. On the other hand, we observe ciliary tip localization of Cp110 in a subset of GRP cilia.

GRP cilia have to be resorbed after LR-asymmetric gene expression is induced, to allow these cells to re-enter the cell cycle and to contribute to other embryonic structures like the somites or the notochord (*Komatsu et al., 2011*; *Shook et al., 2004*). At the GRP, cells ingress from lateral to medial, and cilia with Cp110 at their tips were more frequently found on lateral GRP cells. At ciliary tips, Cp110 may promote axoneme depolymerization via recruitment of Kif24, which was previously shown to interact with Cp110 and to specifically depolymerize centriolar-derived microtubules, but not cytoplasmic microtubule populations (*Kobayashi et al., 2011*; *Kim et al., 2015*).

Interaction of Cp110 with distinct protein complexes was previously proposed (*Tsang and Dynlacht, 2013*) and our data support this idea. CCD-containing proteins are commonly found among centriolar/basal body components, and are thought to regulate pericentriolar material as well as centrioles by acting as a structural lattice and by mediating protein-protein interactions (*Tsang and Dynlacht, 2013*; *Kuhn et al., 2014*; *Salisbury, 2003*). Deletion of Cp110's CCDs prevents efficient cilia inhibition, and decreases binding to centrioles as well as to the distal basal body, but not to rootlets. Therefore, it will be interesting to further dissect Cp110-binding to basal body-associated protein complexes in the future. Our Cp110 deletion constructs might facilitate such dissection using a proteomics approach, as some of them display opposing functions and localization patterns.

Our study also reveals that Cp110 levels need to be precisely controlled for efficient ciliogenesis. In MCCs, *cp110* expression is induced by ciliary transcription factors, and these also regulate the expression of inhibitory miRNAs from the *miR-34/449* family. This co-regulation establishes a gene regulatory module that confers robust *cp110* expression, while preventing excess Cp110 buildup by post-transcriptional regulation. Such regulatory modules might also exist in other ciliated cell types, which express distinct sets of cell type-specific miRNAs (*Walentek et al., 2014*). In the zebrafish embryonic LR-organizer, motile mono-cilia require *miR-129-3p*, which also controls Cp110 levels (*Cao et al., 2012*). Our ChIP-seq data show Foxj1 and RFX2 binding to the *cp110* transcriptional start site, and these transcription factors control motile mono-cilia formation as well (*Choksi et al., 2014*). Therefore, RFX2 and Foxj1 could form a similar module with *cp110* and *miR-129-3p* in the vertebrate embryonic left-right organizer.

Given the importance of Cp110 in ciliogenesis, cell division and pathogenesis, our study contributes important mechanistic insights into the roles of Cp110 during cilia formation and function, which will facilitate further understanding of complex protein networks in cilia-dependent development and disease.

## Materials and methods

### Manipulation of *Xenopus* embryos and constructs used

X. *laevis* eggs were collected and in vitro-fertilized, then cultured and microinjected by standard procedures (*Sater, 2011*). Embryos were injected with Morpholino oligonucleotides (MOs, Gene Tools), mRNAs and DNAs at the two- and four-cell stage using a PicoSpritzer setup in 1/3x Modified Frog Ringer's solution (MR) with 2.5% Ficoll PM 400 (GE Healthcare, #17-0300-50), and were transferred after injection into 1/3x MR containing Gentamycin. Drop size was calibrated to about 7–8 nL per injection. Rhodamine-B dextran (0.5–1.0 mg/mL; Invitrogen, #D1841) or indicated mRNAs were co-injected and used as lineage tracers. *cp110* MO (5'-ACTCTTCATATGGCTCCATGGTCCC-3'; Gene tools) (*Song et al., 2014*) was administered at doses ranging between 17 ng and 60 ng (or 3–7 pmol). mRNAs encoding Centrin4-RFP/CFP (*Antoniades et al., 2014*; *Park et al., 2008*), Clamp-RFP/GFP (*Park et al., 2008*), FAK-GFP (*Antoniades et al., 2014*), Vinculin-GFP (*Antoniades et al., 2014*), Paxillin-GFP (*Antoniades et al., 2014*), GFP-Cp110 in pCS107 and derivatives (this study), GFP-Cep97 in pCS107 (this study), GFP-Cfap20 (gift from BJ Mitchell) were prepared using the Ambion mMessage Machine kit using Sp6 (#AM1340) and diluted to 30–150 ng/µL (240 pg–1.2 ng per injection) for injection into embryos. *Xenopus tropicalis gfp-cep97* cDNA was derived from IMAGE clone #780092 and subcloned using BamH1 and Sal1 enzymes (New England Biolabs) after amplification using following primers:

Cep97-BamH1-forward: AAAAAAGGATCCATGGCAGTGGCACATTTG
Cep97-Sal1-reverse: AAAAAAGTCGACTTAAAGGACTAATTCTGGCTGTG.

*Xenopus tropicalis cp110* cDNA was derived from a clone matching BC167469 obtained from Thermo Scientific (#MXT1765-202715711). The Xt *cp110* reference sequence (Gene ID: 100170501)

was corrected by linking to the *Xenopus tropicalis* genome by NCBI on 23. June 2015. *Gfp-cp110-fs* and g*fp-cp110-fsΔmiR-34/449* were generated from the same clone and subcloned into the pCS107 expression vector, which was digested with Sph1 (New England Biolabs, #R0182S) and re-ligated to remove the miR binding site, as previously described (*Song et al., 2014*). DNAs were purified using the PureYield Midiprep kit (Promega, Madison, WI, USA; #A2495), and were injected at 1–2 ng/µl, as previously described (*Song et al., 2014*; *Walentek et al., 2012*; *Walentek et al., 2015*; *Walentek et al., 2013*). Subcloning was performed using BamH1 and EcoR1 (New England Biolabs, #R0101T; #R0136T) restriction enzymes and the following primers (shown 5′ to 3′):

  Cp110 forward AAAAAAGGATCC ATGGAGCCATATGAAGAATTTTATAAG;
  Cp110 reverse GCTGAAGAATTCTGTTCTCTGAG;
  GFP forward AAAAAAGGATCCATGGTGAGCAAGGGCGAGGAGCTGTTC;
  GFP reverse AAAAAAGGATCCCTTGTACAGCTCGTCCATGCCGAGAGTG;
  FLAG forward AAAAAAGGATCCATGGATTACAAGGATGA;
  FLAG reverse AAAAAAGGATCCTTTATCGTCATCATCTTT.

Cp110 deletion constructs were cloned using the NEB Q5 Site-Directed Mutagenesis Kit (#E0554S). All constructs were verified by sequencing. For in silico translation, Transeq (http://www.ebi.ac.uk/Tools/st/emboss_transeq) was used. For prediction of coiled-coil domain clusters COILS (http://www.ch.embnet.org/software/COILS_form.html) was used. miRNA target sites were predicted using TargetScan (http://www.targetscan.org/vert_71/) and RNA22 (https://cm.jefferson.edu/rna22/).

## Statistical evaluation

Statistical evaluation of experimental data was performed using chi-squared tests (http://www.physics.csbsju.edu/stats/contingency.html) for all data depicted by stacked bar-graphs, or Wilcoxon sum of ranks (Mann-Whitney) tests (http://www.fon.hum.uva.nl/Service/Statistics/Wilcoxon_Test.html) for all data depicted by box-plots (the whiskers (95%) of the box (50%) plots extend to maximal 1.5x IQR, and outliers are displayed as circles).

## Immunofluorescent staining and sample preparation

For *Xenopus* antibody staining, immunofluorescence was performed on whole-mount embryos fixed at embryonic stages 30–33 (mucociliary MCCs), stage 20 (apical basal body transport in MCCs) or stages 16/17 (left-right cilia) in 4% paraformaldehyde at 4°C over night. Embryos were washed 3x 15 min with PBS, then 2x 30 min in PBST (0.1% Triton X-100 in PBS), and were blocked in PBST-CAS (90% PBS containing 0.1% Triton X-100, 10% CAS Blocking; ThermoFischer #00–8120) for 1 hr at RT. Primary and secondary antibodies were applied in 100% CAS Blocking over night at 4°C. Actin staining was performed by incubation (30–60 min at room temperature) with AlexaFluor 488- or 647-labeled Phalloidin (1:40; Molecular Probes #A12379 and #A22287).

For immunofluorescence staining of human airway epithelial cells (HAECs), primary human cells were grown using standard air-liquid interface (ALI) culture by the Walter E. Finkbeiner laboratory at University of San Francisco for 28 days (*Fulcher et al., 2005*). Cells were fixed in 4% PFA or Dent's 80% methanol (EMD, #MX0485P-4) with 20% DMSO (Fisher Scientific, #BP231-100) for 24 hr at −20°C and processed for staining as described for *Xenopus* samples.

For immunofluorescence staining on cryosections, whole tracheas of adult wildtype Black 6 (C57BL/6J) mice were fixed overnight at −20°C in Dent's. Tracheas were embedded in (1:1) 20% Sucrose and O.C.T. compound (Tissue-Tek, #4583) and sectioned with MICROM HM 550 at −18°C at a thickness of 12 µm. Slides were washed in PBS (3 × 15 min), blocked (1 hr at room temperature) in PBST-CAS, and incubated (overnight at 4°C) with primary antibodies. Slides then were washed three times in PBST, and incubated (2 hr at room temperature or over night at 4°C) with secondary antibody. Slides were counterstained using DAPI (4′,6-Diamidino-2-Phenylindole, Dihydrochloride; Molecular Probes, #D1306). Slides were mounted with VECTASHIELD mounting medium (Vector Laboratories, #H-1000-10).

Primary antibodies: mouse monoclonal anti-Acetylated-α−tubulin (in Xl, Mm, Hs; 1:700; Sigma #T6793), rabbit polyclonal anti-Cp110 (in Mm, Hs; 1:200; Proteintech #12780-1-AP), mouse anti-Centrin1 (in Hs; 1:200; clone 20H5 EMD Milipore #04–1624). Secondary antibodies (1:250): AlexaFluor 555-labeled goat anti-mouse antibody (Molecular Probes #A21422), AlexaFluor 555-labeled goat

anti-rabbit antibody (Molecular Probes #A21428), AlexaFlour 488-labeled goat anti-rabbit antibody (Molecular Probes #R37116) and AlexaFluor 405-labeled goat anti-mouse antibody (Molecular Probes #A31553). Z-stack analysis and processing were performed using ImageJ (*Schindelin et al., 2012*). Lateral projections were computed using Zeiss ZEN software. All confocal imaging was performed using a Zeiss LSM700.

3D-SIM imaging was performed on a Zeiss Elyra SR.1 (3 angels) on samples embedded in Pro-Long Gold (Thermo Fisher #P36930) for 48 hr and high-precision cover slips (Zeiss #474030-9010-000) were used. 3D-SIM reconstruction was performed using Zeiss Zen software and calibration using multicolor fluorescent beads was performed prior to channel alignment.

## Co-immunoprecipitation and western blotting

Embryos were injected 4x at the four-cell stage and animal caps were prepared at stage 9. At stage 28, 15 caps per condition were pooled in 100 µl TNMEN-150 lysis buffer (150 mM NaCl, 1 mM EDTA, 2 mM MgCl2, 0.1% Nonidet-P40, 50 mM This pH8.0, 1x Roche cOmplete (#04693116001)). Co-Immunoprecipitation was performed following standard protocol, using 10 µl magnetic beads (Dynabeads M-280 Sheep Anti-Mouse IgG; #11202D) and 0.4 µl monoclonal mouse anti-FLAG antibody (Sigma; #F3165) per sample for 2 hr at 4°C. 10% of sample was removed prior to treatment with antibody/magnetic beads (input), and 30 µl of sample was removed after the treatment (supernatant). SDS-Page and Western blotting were performed using standard procedures using a 10% separating gel, Milipore Immobilon-FL PVDF membrane (#IPFL00010), TBS containing 0.1% Tween-20 (TBSw) for washing, TBSw plus 5% non-fat dry milk for blocking. FLAG-/GFP-tagged proteins were detected using monoclonal mouse anti-FLAG antibody (1:2000, Sigma; #F3165), polyclonal rabbit anti-GFP antibody (1:2000, Abcam; #ab290), anti-rabbit/-mouse HRP conjugated secondary antibodies (1:5000, Bio-Rad; Goat anti-Rabbit IgG #1706515 and Goat anti-Mouse IgG #1706516), Western Lightning Plus-ECL (Perkin Elmer; #NEL103E001EA), and Amersham Hyperfilm ECL (GE Healthcare Life Sciences; #28906836).

## Imaging of extracellular fluid flow

For imaging of extracellular fluid flow, control and manipulated stage 32 embryos were anesthetized (Benzocaine, Sigma #E1501) and exposed to latex beads (FluoSpheres carboxylate-modified microspheres, 0.5 µm, red fluorescence [580/605], 2% solids, Invitrogen #F-8812; diluted to 0.04% in 1/3 x MR) in a sealed flow chamber. Time-lapse movies (10 s / 60 frames per s) were recorded using epifluorescent illumination at 20x magnification on a Zeiss Axioskop 2 in combination with a high-speed GX-1Memrecam (NACImage Technology) and processed in ImageJ for brightness/contrast. Particle linking, tracking and quantification of extracellular fluid flow velocities was performed as previously described using the Particle Tracker plugin for ImageJ and a customized R-script (*Hagenlocher et al., 2013*). Frames were reduced to 1/3 to create a 10 s movie and play rate was adjusted to 20 frames per second. Supplemental *Video 1* plays at 1x speed.

## Imaging of MCC cilia motility

For imaging of MCC cilia motility, control and manipulated stage 30 embryos were anesthetized (Benzocaine, Sigma #E1501) and imaged at a rate of 30 frames per second using a Nikon Eclipse Ti inverted confocal microscope equipped with a resonance scanner and NIS Elements Confocal software, as described (*Turk et al., 2015*). Maximum intensity projections were generated in ImageJ to visualize ciliary beating directionality in stills. The movies (*Videos 2,3*) were cropped and adjusted for brightness/contrast in ImageJ, frames were reduced to 1/3 and the play rate was adjusted to 10 frames per second. Movies play at 1x speed. *Video 2* depicts single optical plane sections through the apical-basal axis of MCCs (lateral view of the MCC). *Video 3* depicts single optical plane sections through the ciliary tuft of MCCs (top view on the MCC).

## Quantification of basal body numbers in deep cytoplasm

Confocal z-stacks from controls, *cp110* morphants, and rescued *cp110* morphants were analyzed for the presence of apically localized basal bodies and basal bodies that remained deep in the cell. Apically localized basal bodies were defined as present in the first (apical) 4 z-sections containing Centrin4-CFP signal (=1.82 µm); deep localized basal bodies were defined as Centrin4-CFP signals

located below 1.82 mm. Next, we used the 3D Objects Counter plugin in ImageJ to quantify basal bodies that remained deep in the cytoplasm. These automatic quantifications were then inspected and corrected in cases where objects other than basal bodies were detected (assigned object count = 0), and where two or more basal bodies were counted as one (assigned object count = 2 or more).

## Analysis of basal body FAK-GFP, Vinculin-GFP, and Paxillin-GFP localization

Imaging was performed using the same settings within individual experiments on embryos which were injected with equal amounts of mRNAs. Four apical optical sections (most apical section determined by appearance of Centrin4-CFP) were chosen and processed using ImageJ to adjust brightness/contrast and to generate maximum intensity projections. Brightness for depicted images was further adjusted to match maximum intensity levels at lateral membrane foci to account for variation in expression levels. For quantification of FAK-GFP/Centrin4-CFP ratio all z-planes were used, and the same adjustment of brightness/contrast was performed in ImageJ for samples from the same experiment. Maximum intensity projections were generated and the basal body-containing central region (without lateral membranes) was chosen as the region of interest (ROI) to analyze gray values for each channel separately using ImageJ. Intensity ratios were calculated and normalized using the average intensities in controls (set to 1) for each experiment.

## Analysis of left-right axis development and neural gene expression

For analysis of left-right axis development and ciliation of the GRP, embryos were injected two times into the dorsal marginal zone at the four-cell stage (*Walentek et al., 2012*). For analysis of neural gene expression embryos were injected dorsal-animally at the four-cell stage. For GRP analysis, embryos were stained as described above. For quantification of ciliation rates, cilia length and polarization, central GRP areas were analyzed using ImageJ and R as previously described (*Walentek et al., 2012*). In situ hybridization was performed using standard procedures (*Harland, 1991*) after fixation in MEMFA (100 mM MOPS pH 7.4, 2 mM EGTA, 1 mM MgSO4, 3.7% (v/v) formaldehyde) for 2 hr at room temperature. A Digoxigenin-labeled (Roche, #11209256910) antisense probe was generated using Sp6 or T7 RNA polymerase (Promega, #P1085; #P2075) from plasmids encoding *pitx2c* (*Schweickert et al., 2001*), *nkx2.2* (*Dessaud et al., 2008*), and *pax6* (plasmid matching NP_001079413.1). Embryos were bleached after staining by standard procedures to remove pigment, and for neural gene expression embryos were imaged in 1: 2 (vol: vol) benzylbenzoate: benzyl alcohol (BB:BA).

## Quantitative RT-PCR

*Xenopus* mucociliary organoids were generated from animal caps, dissected in 1x Modified Barth's Saline from stage 9 embryos, which were either uninjected or injected four times with the indicated constructs (mRNAs or MOs). Explants were cultured in 0.5xModified Barth's Saline until unmanipulated control embryos reached indicated stages. Total RNA was isolated by Trizol (Invitrogen, #15596) from 15 explants per condition and experiment. For RT-qPCR, cDNA was generated from total RNA extracts using iScript Reverse Transcription Supermix (BioRad; #170–8840); and the following qPCR primers were used: *Foxj1*-F: CCAGTGATAGCAAAAGAGGT, and *Foxj1*-R: GCCATGTTC TCCTAATGGAT; *Cp110*-F: AGCCAGAATCCAAGTAAAGG, and *Cp110*-R: CTTGCTTCTTTTCAG-CAGTC; *EF1a*-F: CCCTGCTGGAAGCTCTTGAC, and *EF1a*-R: GGACACCAGTCTCCACACGA; *ODC*-F: GGGCTGGATCGTATCGTAGA, and *ODC*-R: TGCCAGTGTGGTCTTGACAT. Reactions were performed on a BioRad CFX96 Real-Time System C1000 Touch.

For miRNA quantitation, Trizol prepared total RNA was poly (A)-tailed by Poly (A) Polymerase (Epicentre, #PAP5104H). Poly (A)-tailed small RNA was reverse transcribed into small RNA cDNA with SuperScript III reverse transcriptase (Invitrogen, #18080) using miRNA RT primer (CGAATTC TAGAGCTCGAGGCAGGCGACATGGCTGGCTAGTTAAGCTTGGTACCGAGCTCGGATCCACTAG TCCTTTTTTTTTTTTTTTTTTTTTTTTTTTVN). (V is A, G or C; N is A, G, C or T). TaqMan-based qPCR was subsequently performed on a 7900HT fast real-time PCR system (Applied Biosystems). The U6 snRNA was used as the endogenous control for miRNA real time qPCR analyses. Universal TaqMan probe, CTCGGATCCACTAGTC; Universal reverse primer, CGAATTCTAGAGCTCGAGGCAG. The following forward primers, specific for each small RNA, were used: ATGTGAAGCGTTCCATATGA;

*miR-34a*: TGGCAGTGTCTTAGCTGGTTGTT; *miR-34b*: CAGGCAGTGTAGTTAGCTGATTG; *miR449c*: TGCACTTGCTAGCTGGCTGT.

## RNA-sequencing and chromatin immunoprecipitation and DNA-sequencing

RNA-seq libraries: RNAs were isolated by the proteinase K method followed by phenol-chloroform extractions, lithium precipitation, and treatment with RNase-free DNase and a second series of phenol-chloroform extractions and ethanol precipitation. RNAseq libraries were then constructed (Illumina TruSeq v2; #RS-122-2001) and sequenced on an Illumina platform. RNAseq reads are deposited at NCBI (GSE76342).

RNAseq informatics: Sequenced reads from this study or (*Chung et al., 2014*; *Ma et al., 2014*) were aligned to the *X. laevis* transcriptome, MayBall version with RNA-STAR (*Dobin et al., 2013*) and then counted with eXpress. DESeq (*Roberts et al., 2013*) was used to estimate dispersion and test differential expression using rounded effective counts from eXpress. Changes in expression were visualized in R with beanplot (https://cran.r-project.org/web/packages/beanplot/beanplot.pdf), and to visualize RNAseq reads in a genomic context they were mapped to genome version 9.1 with bwa mem (*Li and Durbin, 2009*) and loaded as bigWig tracks into the Integrative Genomics Viewer browser (*Thorvaldsdottir et al., 2013*).

ChIPseq libraries: Samples were prepared for ChIP using methods described (*Ma et al., 2014*) with the following modifications: About 250 animal caps for transcription factors or 100 caps for histone modifications were fixed for 30 min in 1% formaldehyde, and chromatin was sheared on a Bio-Ruptor (30 min; 30 s on and 2 min off at the highest power setting). Tagged proteins with associated chromatin were immunoprecipitated with antibodies directed against GFP (Invitrogen; #A11122 lot #1296649), FLAG (Sigma; #F1804), H3K4me3 (Active Motif; #39159 lot #01609004), or H3K27ac (Abcam; #ab4729, lot #GR71158-2). DNA fragments were then polished (New England Biolabs, end repair module; #E6050S), adenylated (New England Biolabs, Klenow fragment 3′–5′ exo- and da-tailing buffer), ligated to standard Illumina indexed adapters (Illumina TruSeq v2; #RS-122-2001), PCR-amplified (New England Biolabs, Phusion #M0530S or Q5 #M0491S, 16 cycles), and sequenced on an Illumina platform. ChIPseq reads are deposited at NCBI (pending).

ChIPseq informatics: ChIP-seq reads from this study or from (*Chung et al., 2014*; *Ma et al., 2014*) were mapped to *X. laevis* v9.1 with bwa mem, peaks called with HOMER (*Heinz et al., 2010*) using input as background and loaded as bigWig tracks into the Integrative Genomics Viewer browser. Peak positions were annotated relative to known exons (Mayball gene models).

## Sample size and analysis

Sample sizes for all experiments were chosen based on previous experiences and performed in embryos derived from at least two different females. No randomization or blinding was applied.

## Acknowledgements

We thank M Blum, BJ Mitchell, A Schweickert, PA Skourides and JB Wallingford and their labs for sharing of constructs and unpublished data, P Lishko for using GX-1 Memrecam, and the Wallingford lab for using Nikon Eclipse Ti microscope. We thank R Song, L He for discussions, help with miR qPCR and miR target analysis; and C Boecking, L Zlock, W Finkbeiner for HAECs (Cystic Fibrosis Cell Models Core funded by NIH DK072517 and Cystic Fibrosis Foundation DR613-CR11). Work in the Kintner lab was funded by NIH grant 5R01GM076507 to CK. We thank HL Aaron and J-Y Lee (Berkeley Imaging Center) for imaging support, and D Schichnes (Berkeley Biological Imaging Facility) for help with 3D-SIM (supported by NIH, Health S10 program 1S10OD018136-01). The National Xenopus Resource (RRID:SCR_013731) and Xenbase (RRID:SCR_003280) were continuously used throughout the project. We thank C Exner for careful reading of the manuscript and Edivinia and Elleanor Pangilinan for expert technical help. PW was funded by the Deutsche Forschungsgemeinschaft (DFG, Wa 3365/1-1) and NIH-NHLBI (K99HL127275). *Xenopus* work in the Harland laboratory was funded by NIH grant GM42341 to RMH.

## Additional information

### Funding

| Funder | Grant reference number | Author |
|---|---|---|
| Deutsche Forschungsge-meinschaft | Wa 3365/1-1 | Peter Walentek |
| National Heart, Lung, and Blood Institute | K99HL127275 | Peter Walentek |
| National Institute of General Medical Sciences | GM42341 | Richard M Harland |
| National Institute of General Medical Sciences | GM076507 | Christopher Kintner |

The funders had no role in study design, data collection and interpretation, or the decision to submit the work for publication.

### Author contributions

PW, Designed and performed experiments, Interpreted the data, Wrote the manuscript; IKQ, CK, Contributed RNA-seq and ChIP-seq data and analysis; DIS, UKS, Contributed to molecular cloning, in situ hybridization, sectioning and immunofluorescence staining; RMH, Contributed to experimental design, Interpretation of data, Manuscript preparation

### Author ORCIDs

Peter Walentek, http://orcid.org/0000-0002-2332-6068
Ian K Quigley, http://orcid.org/0000-0003-0075-8324
Richard M Harland, http://orcid.org/0000-0001-8247-4880

### Ethics

Animal experimentation: This work was done with approval of University of California, Berkeley's Animal Care and Use Committee. University of California, Berkeley's assurance number is A3084-01, and is on file at the National Institutes of Health Office of Laboratory Animal Welfare.

## Additional files

### Major datasets

The following dataset was generated:

| Author(s) | Year | Dataset title | Dataset URL | Database, license, and accessibility information |
|---|---|---|---|---|
| Quigley IK, Kintner C | 2015 | RNAseq profiling of multiciliated cells | http://www.ncbi.nlm.nih.gov/geo/query/acc.cgi?acc=GSE76342 | Publicly available at the NCBI Gene Expression Omnibus (Accession no: GSE76342) |

The following previously published datasets were used:

| Author(s) | Year | Dataset title | Dataset URL | Database, license, and accessibility information |
|---|---|---|---|---|
| Ma L, Quigley IK, Kintner C | 2014 | Multicilin drives centriole biogenesis via E2f proteins | http://www.ncbi.nlm.nih.gov/geo/query/acc.cgi?acc=GSE59309 | Publicly available at the NCBI Gene Expression Omnibus (Accession no: GSE59309) |

| Chung M, Kwon T, Gupta R, Baker JC, Marcotte EM, Wallingford JB | 2014 | Coordinated genomic control of ciliogenesis and cell movement by RFX2 | http://www.ncbi.nlm.nih.gov/geo/query/acc.cgi?acc=GSE50593 | Publicly available at the NCBI Gene Expression Omnibus (Accession no: GSE50593) |

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
