## [Decision Letter]

Thank you for submitting your article "Ciliary transcription factors and miRNAs precisely regulate Cp110 levels at basal bodies required for ciliogenesis" for consideration by *eLife*. Your article has been reviewed by three peer reviewers, and the evaluation has been overseen by Janet Rossant as the Senior and Reviewing Editor. The reviewers have opted to remain anonymous.

Summary:

Using *Xenopus* embryos, the authors show that optimal Cp110 levels are required for cilia formation and that Cp110 is needed for ciliary adhesion complex formation. The ciliation defects in *cp110* morphants were initially reported by the authors in (Song et al., 2014), and this study extends these observations to examine the mechanism of Cp110 in promoting ciliogenesis, as compared to its well-studied role in limiting cilium assembly. These studies also confirm certain recent findings in knock-out mice. The most novel data pertain to a role for Cp110 in regulating actin pathways and the ciliary adhesion complex. However, the latter data need substantial corroboration as described below.

Essential revisions:

The reviewers have discussed the reviews with one another extensively and have concluded that the paper as it stands lacks some important data, particularly with regard to the localization of the endogenous components of the complexes in which Cp110 is proposed to be engaged. A significant amount of additional data is required for us to reconsider the paper, including controls for the main experiments presented in terms of localisation and CP110 interaction with the ciliary adhesion complex, FAK and the actin cytoskeleton. The reviewers felt that you might be able to address these issues within the *eLife* two month window for revisions, but only if you have immediate access to appropriate antibodies to carry out the analysis requested. After reading the reviews, please let us know if you feel you can address the comments with new data in a timely manner – direct email discussion with Dr Rossant is simplest.

The full reviews are provided below to help you consider whether the revisions requested can be achieved.

*Reviewer #1:*

Using *Xenopus* embryos, the authors show that optimal Cp110 levels are required for cilia formation and that Cp110 is needed for ciliary adhesion complex formation. In addition, analysis of Cp110 mutants revealed distinct functions for different Cp110 domains. The ciliation defects in *cp110* morphants were initially reported by the authors in (Song et al., 2014), and this study extends these observations to examine the mechanism of Cp110 in promoting ciliogenesis, as compared to its well-studied role in limiting cilium assembly. These studies also confirm certain recent findings in knock-out mice. The most novel data pertain to a role for Cp110 in regulating actin pathways and the ciliary adhesion complex. However, the latter data need substantial corroboration as described below.

1) The authors conclude that Cp110 could promote cilia assembly by facilitating formation of the ciliary adhesion (CA) complex, which was previously shown to link basal bodies and ciliary rootlets to the apical actin network. Because Cp110 localizes to both the basal body and tip of rootlets, does Cp110 MO also affect rootlet structure (e.g., length)? In Figure 4 and Figure 4—figure supplement 1, the localization of ciliary adhesion complex around the basal bodies (possibly the rootlet region) seems to be more dramatically affected than at the basal bodies. Does Cp110 loss lead to overall reduction of CA proteins or preferential disappearance of CA proteins at the tip of rootlet? Triple staining of CA proteins, basal bodies and rootlets may help.

2) Related to the above point: A major conclusion is that Cp110 needs to be kept at optimal levels. In normal embryos, Cp110 and ciliary adhesion complex levels differ among basal bodies at anterior and posterior aspects. How does this asymmetric distribution correlate with the role of Cp110 in ciliogenesis? Basal bodies with lower Cp110 levels are also likely to have lower levels of ciliary adhesion proteins that contribute to basal body docking to actin, so does docking of these basal bodies with lower Cp110 differ as compared to those with higher Cp110 levels? Also, do cilia that emanate from basal bodies with higher and lower Cp110 levels differ? Does overexpression of Cp110 recruit excess CA proteins to the basal bodies/rootlets?

3) Most importantly, data are currently not sufficiently convincing to support a major conclusion, namely, the existence of the Cp110-FAK interaction. In particular, biochemical support for the observation regarding Cp110 interactions with FAK and the actin cytoskeleton is lacking. For example, data in Figure 4 are not of sufficient quality, as only data from over-expression are shown, and adequate negative controls are lacking. Immunoprecipitation of endogenous proteins, ideally using both antibodies, and/or validation of the interaction between Cp110 and another component of the ciliary adhesion complex (vinculin, paxillin) is essential.

4) The authors state (subsection “Cp110 localizes to cilia-forming basal bodies”) that Cp110 plays a role in cilia length control because intermediate levels result in short cilia. Without additional ciliary markers (besides ac-tubulin) and/or ultra-structural studies, this conclusion is not well-supported.

*Reviewer #2:*

The Walentek manuscript details the function of CP110 in regulating cilia formation in MCCs in *Xenopus*. They find that CP110 levels are elaborately regulated by the MCC regulatory network. Depletion of CP110 causes a range of phenotypes including loss of cilia, loss of cilia organization, loss of basal body docking and loss of adhesion complex localization. Overexpression of CP110 has previously been reported to cause ciliogenesis defects, but they further these studies by perform domain mapping to determine the functional domains of CP110 responsible for localization and function at the basal body. Additionally, they perform detailed localization studies and find GFP-CP110 localized adjacent to basal bodies towards the rootlet and weakly at the tip of the rootlet. These results are consistent with a potential role in adhesion complex formation which is disrupted in morphants. CP110 is a very interesting protein and this paper suggests novel previously unappreciated functions for it during ciliogenesis. The manuscript contains a large amount of high quality imaging data and is well written. Clearly the loss of CP110 is detrimental to the function of MCCs. The challenge is determining the primary vs. secondary defects associated with CP110 which I think requires a bit more effort. My main concern is that many of the phenotypes might not be directly attributable to CP110 but to any basal body that has failed to dock.

Comments:

With the rate of videos presented I am not sure one can make too strong a claim about directionality (I don't doubt the overall claim but the data in Figure 1—figure supplement 1 seems a bit subjective to me). While this claim is consistent with the rootlet polarity (1C) it is a bit concerning as one does not know which rootlets actually contain a cilium, as most probably do not give the observed ciliogenesis phenotype. Since rootlets without cilia or with immotile cilia will likely not be polarized, I suspect that any polarity defect is secondary to ciliogenesis and should be presented as such.

Figure 1—figure supplement 1. I could not find a description of what constitutes a mild vs. severe defect in docking and I worry that this data is also a bit subjective. As this phenotype is likely contributing to many (or all) of the other phenotypes it seems critically important.

The claim that Cp110 localizes to "cilia forming basal bodies" seems a bit restrictive, as S3G clearly shows CP110 at non-cilia forming centrioles. Also, the main claim of this manuscript is that CP110 levels are critical, so the overexpression (and therefore localization) of CP110 might not reflect the endogenous situation in regards to localization. This data, while reasonable should be qualified and more clearly stated.

"At the apical membrane, GFP-Cp110 localized between Centrin4-CFP and F-actin (Figure 4—figure supplement 1), supporting the idea that Cp110 might facilitate F-actin binding." I have several concerns with this figure. First off, from my reading of the EM literature there is a complex 3D architecture of actin in MCCs. This analysis appears to be done at the level of the lower basal body / upper rootlet and I am unclear what pool of actin is being analyzed. Second the quality of the image together with the somewhat diffuse actin staining does not seem to me to reflex any consistent relationship between CP110 localization and actin.

"We conclude from these experiments that Cp110 is required for normal formation of ciliary adhesion complexes in MCCs, and suggest that loss of FAK, Vinculin and Paxillin from basal bodies and the apical membrane causes the observed defects in basal body transport, docking and alignment, as well as loss of apical actin." The claim that CP110 somehow regulates the adhesion complex seems concerning. If basal bodies have failed to dock then I am not sure what one would expect to find in regards to the adhesion complex. I am concerned that these adhesion complex phenotypes are completely secondary to docking failure and not that FAK is regulating docking.

"In MCCs, gfp-cp110DCentral induced strong aggregation and an increased number of basal bodies (Figure 5—figure supplement 2), implicating Cp110 in regulation of basal body biogenesis and separation via the MCC-specific deuterosome pathway." This is most certainly an overstatement. Multinucleated cells have more "stuff" and therefore make more cilia, independent of what is causing the cytokinesis defect.

*Reviewer #2 (Additional data files and statistical comments):*

Most of the data is solid. A better description of some of the "subjective" quantifications should be provided. Full images of the IP would help in the interpretation of the rigor of these experiments.

*Reviewer #3:*

Walentek, Harland and coauthors here present a detailed analysis of the impacts of CP110 deficiency and overexpression on ciliogenesis, primarily in *Xenopus*. The regulation of Cp110 levels through miRNAs in ciliogenesis was shown in a recent paper from the authors (Song et al. (2014) Nature 510: 115-120) and the dual roles of Cp110 in ciliogenesis have been demonstrated in the mouse (Yadav et al. (2016) Development 143:1491-1501). To make the current submission a significant advance on this previous work, there are some important control experiments needed.

1) The localizations shown for GFP-Cp110 are inconsistent in Figure 3: Figure 3 shows GFP-CP110 away from the distal end of the basal body, in contrast to its localization in Figure 3' at the distal end and possibly at the rootlet, and apparently at both centrioles in the mono-ciliated cells. As the main topic of the MS. is the importance of CP110 levels, the concern about promiscuous localization of over-expressed CP110 should be addressed. Where is the endogenous protein?

2) Better controls should be shown for the IP experiments in Figure 4. The co-IPs with FAK-GFP and Centrin4-GFP are effectively the negative controls for each other, so they should be included on the same membrane.

3) It is not clear whether Cp110-FS is actually expressed, or whether this was an error in the deposited sequence. The *Xenopus tropicalis* CP110 sequence should be verified by RT-PCR and the correct sequence deposited. The section on the Xt sequence should be rewritten accordingly.

4) The expression levels of the various Cp110 deletion constructs appear different by GFP fluorescence. An immunoblot should be included to show that the different effects are not due to different expression levels or altered protein stability.

5) Arising from point 4., are post-translational regulation of Cp110 levels important in MCCs? Given the published work on the cell cycle regulation of Cp110, some discussion/ clarification of this point for the differentiated cells would be relevant.

[Editors' note: further revisions were requested prior to acceptance, as described below.]

Thank you for resubmitting your work entitled "Ciliary transcription factors and miRNAs precisely regulate Cp110 levels required for ciliary adhesions and ciliogenesis" for further consideration at *eLife*. Your revised article has been favorably evaluated by Janet Rossant as the Senior editor and Reviewing editor, and three reviewers.

The manuscript has been improved and provides important new insights into the roles of Cp110 in ciliogenesis, but there is a remaining issue that needs to be addressed before acceptance, as outlined below:

The main concern remains that the conclusions on the localization of Cp110 are largely based on expression of GFP-CP110, which may be over-expressed relative to endogenous protein, and that they lack quantitation. The authors should present evidence that the unconventional localization is not background and that it is a statistically significant occurrence. It was recognized that the findings of the paper are novel and interesting and should not be held hostage to the lack of all the necessary reagents to confirm native localization and interactions. However, the reviewers felt that the claims of the paper on these issues should be modified and the caveats made clearer.

---

## [Author Response]

*The reviewers have discussed the reviews with one another extensively and have concluded that the paper as it stands lacks some important data, particularly with regard to the localization of the endogenous components of the complexes in which Cp110 is proposed to be engaged. A significant amount of additional data is required for us to reconsider the paper, including controls for the main experiments presented in terms of localisation and CP110 interaction with the ciliary adhesion complex, FAK and the actin cytoskeleton. The reviewers felt that you might be able to address these issues within the eLife two month window for revisions, but only if you have immediate access to appropriate antibodies to carry out the analysis requested. After reading the reviews, please let us know if you feel you can address the comments with new data in a timely manner – direct email discussion with Dr Rossant is simplest.*

*The full reviews are provided below to help you consider whether the revisions requested can be achieved.*

We thank the reviewers and editors for their thoughtful suggestions and critical assessment of our manuscript. We have addressed all of the reviewers’ concerns through clarification and experiments. Most importantly, we now provide additional evidence for the regulation of ciliary adhesion complexes by Cp110. We present new conventional and super-resolution immunofluorescence imaging data on endogenous Cp110 localization at basal bodies in human airway multiciliated cells, which confirms our findings from GFP-Cp110 overexpression.

Furthermore, we show co-localization of Cp110 and FAK at posterior basal bodies, and dosedependent recruitment of FAK to basal bodies and rootlets upon Cp110 overexpression in Xenopus, which supports the idea that different doses of Cp110 have different effects on the cilia. We now also show that defects in basal body alignment and ciliary adhesion localization are not secondary to defects in basal body docking, since we have identified conditions where basal body docking and cilia formation are normal, but alignment is affected. Finally, we show that the differential effects of Cp110 deletion constructs are not primarily caused by differences in expression levels.

*Reviewer #1:*

*Using Xenopus embryos, the authors show that optimal Cp110 levels are required for cilia formation and that Cp110 is needed for ciliary adhesion complex formation. In addition, analysis of Cp110 mutants revealed distinct functions for different Cp110 domains. The ciliation defects in cp110 morphants were initially reported by the authors in (Song et al., 2014), and this study extends these observations to examine the mechanism of Cp110 in promoting ciliogenesis, as compared to its well-studied role in limiting cilium assembly. These studies also confirm certain recent findings in knock-out mice. The most novel data pertain to a role for Cp110 in regulating actin pathways and the ciliary adhesion complex. However, the latter data need substantial corroboration as described below.*

*1) The authors conclude that Cp110 could promote cilia assembly by facilitating formation of the ciliary adhesion (CA) complex, which was previously shown to link basal bodies and ciliary rootlets to the apical actin network. Because Cp110 localizes to both the basal body and tip of rootlets, does Cp110 MO also affect rootlet structure (e.g., length)? In Figure 4 and Figure 4—figure supplement 1, the localization of ciliary adhesion complex around the basal bodies (possibly the rootlet region) seems to be more dramatically affected than at the basal bodies. Does Cp110 loss lead to overall reduction of CA proteins or preferential disappearance of CA proteins at the tip of rootlet? Triple staining of CA proteins, basal bodies and rootlets may help.*

We agree with the reviewer that loss of ciliary adhesion complex proteins appears more dramatic at the rootlet than at the basal body, although clearly both populations are affected. So, to address this we provide additional experiments in the revised manuscript, in which we demonstrate the dose-dependent effects of *cp110* knockdown on MCCs (Figure 1—figure supplement 1). At low concentrations, *cp110* MO caused disruption of basal body alignment and mild apical actin defects, but without major impact on ciliation or apical basal body transport/docking or rootlet formation. These data support the conclusion that the rootlet complex of ciliary adhesion proteins is most sensitive to Cp110 deficiency, possibly because Cp110 levels are relatively low at the rootlet tip (Figure 3’).

We also provide a triple staining in controls and *cp110* morphants using Centrin-CFP to mark the basal body, Clamp-RFP as rootlet marker, and FAK-GFP for ciliary adhesions, as suggested by the reviewer. We now show in Figure 4—figure supplement 1 that Clamp and Centrin4 localize to rootlets/basal bodies in both controls and *cp110* morphant MCCs, while FAK is selectively and strongly reduced in *cp110* morphants. These results indicate that loss of the rootlet is unlikely the primary cause for loss of ciliary adhesion complexes and failure of ciliogenesis in Cp110-deficient MCCs. Further analysis of apical versus deep basal bodies in *cp110* morphant MCCs (Figure 4—figure supplement 1’) revealed that Clamp staining in apically localized basal bodies was similar to controls, while Clamp concentrations at basal bodies that remained deep in the cytoplasm were reduced, indicating possible defects in the most significantly affected basal bodies.

Our results are in line with Yadav et al. (Development, 2016, 143, 1491-1501), who show that Rootletin still localizes to basal bodies/rootlets in Cp110-/- mice. Furthermore, Yang et al. (2005, Mol Cell Biol. 25(10):4129-37) previously reported that rootlets are not required for basal body docking and ciliogenesis in the mouse. We refer to these observations in the third paragraph of the subsection “Cp110 is required for ciliary adhesion complex formation”.

Taken together, we conclude that loss of ciliary adhesion complexes and cilia formation is not primarily caused by a loss of the rootlet, and that ciliary adhesion complexes are strongly reduced in Cp110-deficient basal bodies even when the rootlet marker Clamp is still present.

*2) Related to the above point: A major conclusion is that Cp110 needs to be kept at optimal levels. In normal embryos, Cp110 and ciliary adhesion complex levels differ among basal bodies at anterior and posterior aspects. How does this asymmetric distribution correlate with the role of Cp110 in ciliogenesis?*

Our data suggest that *cellular* Cp110 levels need to be optimal to allow for successful ciliogenesis. The finding that levels of ciliary adhesion complexes and Cp110 also vary along the axis of polarization is an interesting observation, but at this point we can only speculate about the mechanism of asymmetric distribution as well as about the functional consequences of this asymmetry. It could be related to the mechanism by which tissue-level planar cell polarity signaling (e.g. via Frizzled6 and Vangl1) is transduced to the polarity of basal bodies in MCCs. Vladar and colleagues (2015, Methods in Cell Biology, 127, 37-54) have proposed that in the trachea – where cilia beat along the proximal-distal axis – proximally localized Frizzled connects to the “first row” of basal bodies via microtubules. This is thought to polarize these basal bodies, which pass this polarity on to the remaining basal bodies within the cell via the apical and sub-apical actin networks. Therefore, the differences in levels of ciliary adhesion complex proteins and Cp110 along this axis might reflect the strength of polarization and mechanical coupling. Such a reasoning is also discussed by Antoniades et al. (2014, Dev Cell 28, 70-80).

In fact, we do observe this polarity more frequently in earlier stages, possibly during initial polarization, than in later stages. This suggests that additional ciliary adhesion proteins and Cp110 could be recruited during later steps of polarization to further stabilize basal body orientation, possibly in a similar manner as suggested for Bbof1 (Chien et al. 2013, Development Aug, 140(16):3468-77).

*Basal bodies with lower Cp110 levels are also likely to have lower levels of ciliary adhesion proteins that contribute to basal body docking to actin, so does docking of these basal bodies with lower Cp110 differ as compared to those with higher Cp110 levels?*

In the revised manuscript we present data on *cp110* MO-dose dependent phenotypes in MCCs (Figure 1—figure supplement 1). Low *cp110* MO concentrations lead to largely normal docking of basal bodies and successful ciliogenesis, while interfering selectively with proper basal body alignment. With increasing concentrations, *cp110* MO causes increasingly severe basal body transport and docking defects with the result that cilia are not formed and basal bodies remain deep in the cytoplasm. All these processes require ciliary adhesion complexes, and a similar dose-dependency was reported for FAK knockdown in MCCs (Antoniades et al. 2014, Dev Cell 28, 70-80). Therefore, this data supports our hypothesis that a fairly specific (low) dose of Cp110 is required for normal ciliary adhesion complex formation and function, while excess levels of Cp110 recruit more ciliary adhesions to basal bodies (Figure 4), but simultaneously prevent cilia formation, likely by distal end capping(Kobayashi et al. Centriolar Kinesin Kif24 Interacts with CP110 to Remodel Microtubules and Regulate Ciliogenesis. Cell 145, 914–925 [2011]; Spektoret al. Cep97 and CP110 suppress a cilia assembly program. Cell 130, 678–90 [2007]).

*Also, do cilia that emanate from basal bodies with higher and lower Cp110 levels differ?*

Yes, we do observe a reduction in length of GRP cilia with more Cp110 (Figure 3—figure supplement 2’). We address this in the Results subsection “Cp110 localizes to cilia-forming basal bodies” as well as in the discussion on Cp110’s possible role in cilia length control and resorption (Discussion, fourth paragraph).

*Does overexpression of Cp110 recruit excess CA proteins to the basal bodies/rootlets?*

We thank the reviewer for this suggestion. We performed the experiments and we do see a dose-dependent increase in FAK-GFP localization upon *cp110* overexpression. The results are now shown in Figure 4, and mentioned in the Results subsection “Cp110 is required for ciliary adhesion complex formation” as well as in the second paragraph of the Discussion.

*3) Most importantly, data are currently not sufficiently convincing to support a major conclusion, namely, the existence of the Cp110-FAK interaction. In particular, biochemical support for the observation regarding Cp110 interactions with FAK and the actin cytoskeleton is lacking. For example, data in Figure 4 are not of sufficient quality, as only data from over-expression are shown, and adequate negative controls are lacking. Immunoprecipitation of endogenous proteins, ideally using both antibodies, and/or validation of the interaction between Cp110 and another component of the ciliary adhesion complex (vinculin, paxillin) is essential.*

We agree with the reviewer that Co-IP of endogenous protein would strongly support our experimental findings and the Co-IP data from overexpressed Flag-tagged Cp110. We have tried two commercially available anti-human-Cp110 antibodies and used human airway epithelial cell cultures as our target for immunoprecipitation. Unfortunately, despite obtaining effective ciliation in these cultures over the four week period required, we were limited in the amount of material and could not convincingly Co-IP associated proteins, including endogenous FAK. Nevertheless, our other observations show that the proteins are associated in a functional complex, even though we cannot make a statement that they are in direct contact. The finding that they do participate in the functional regulation of ciliary adhesions as well as basal body interactions with Actin is one of the significant insights coming from the work.

As suggested by reviewers two and three, we now show the full membranes for the Co-IP of overexpressed Cp110 in *Xenopus* in Figure 4—figure supplement 2. In Figure 4, we also show the uninjected control lane. In the revised manuscript, we clearly point out the differences between Co-IPs on endogenous and overexpressed Cp110.

Importantly, however, we provide additional conventional and super-resolution immunofluorescence data of endogenous Cp110 in human airway multiciliated cells, which confirms Cp110 localization at cilia-forming basal bodies in the same way as observed with overexpressed GFP-Cp110 in *Xenopus*. Furthermore, we show co-localization of GFP-Cp110 and FAK-mKate (Figure 4), and that FAK-GFP localization to basal bodies correlates with Cp110 levels in loss- and gain-of-function experiments (Figure 4).

Collectively, these data support our main conclusion that Cp110 is required for ciliary adhesion complex formation/recruitment to basal bodies. Nevertheless, it remains to be demonstrated if the influence of Cp110 on ciliary adhesion complex proteins is via direct interaction or if it involves additional protein complexes, which we will address in future unbiased proteomics studies.

*4) The authors state (subsection “Cp110 localizes to cilia-forming basal bodies”) that Cp110 plays a role in cilia length control because intermediate levels result in short cilia. Without additional ciliary markers (besides ac-tubulin) and/or ultra-structural studies, this conclusion is not well-supported.*

We appreciate the comment and agree with the reviewer that our rather short statement in the initial manuscript might have led to the impression we are overstating this correlative finding.

We provide additional data in the revised manuscript that support our hypothesis that Cp110 might be involved in ciliary length control/cilia resorption. In a subset of embryos, we observe frequent ciliary tip localization of Cp110 in monociliated Gastrocoel Roof Plate cells (GRP cells of the left-right organizer/node equivalent), which is now shown in Figure 3—figure supplement 2. This observation suggests that Cp110 could be directly involved in cilia length regulation. We think that Cp110 at the ciliary tip could mediate axoneme depolymerization and cilia resorption for the following reasons:

1) Ciliated cells at the left-right organizer are a transient phenomenon during development. After the left-right axis is induced by cilia-driven leftward flow, cells retract their cilia in a coordinated manner which allows re-entry into the cell cycle (Komatsu et al. Development, 2011, Sep;138(18):3915-20) and ingression into the somites, notochord and hypochord in *Xenopus* (Shook et al. Developmental Biology, 2004, 270:163-185). The observation that we find more abundant ciliary tip localization of Cp110 in older embryos is consistent with the idea that Cp110 accumulates in tips of cilia that are actively shortening. Furthermore, we see Cp110 more frequently at ciliary tips at the lateral edges of the GRP, which is in line with the finding that GRP cells ingress from lateral to medial in *Xenopus laevis* (Shook et al. Developmental Biology, 2004, 270:163-185).

2) Cp110 was reported to be part of a cilia suppression complex, which acts via recruitment of negative regulators of tubulin polymerization, i.e. Kif24 (which was shown to specifically depolymerize centriolar microtubules, but not cytoplasmic microtubules; please see Kobayashi et al. Cell, 2011, 145(6):914-25). Therefore, Cp110 could be translocated to the ciliary tip together with Kif24 in order to promote active axoneme depolymerization necessary for cilia resorption.

In the revised manuscript, we now discuss these findings and our hypothesis in the fourth paragraph of the Discussion.

*Reviewer #2:*

*With the rate of videos presented I am not sure one can make too strong a claim about directionality (I don't doubt the overall claim but the data in Figure 1—figure supplement 1 seems a bit subjective to me). While this claim is consistent with the rootlet polarity (1C) it is a bit concerning as one does not know which rootlets actually contain a cilium, as most probably do not give the observed ciliogenesis phenotype. Since rootlets without cilia or with immotile cilia will likely not be polarized, I suspect that any polarity defect is secondary to ciliogenesis and should be presented as such.*

We agree with the reviewer and this is certainly true for *cp110* morphants injected with high doses of *cp110* MO. Our additional data argue that polarity defects can occur in the absence of severe ciliogenesis defects in *cp110* morphants injected with low doses of *cp110* MO. In revised Figure 1—figure supplement 1 and D we now show the dose-dependent effects on ciliation, basal body polarization, basal body localization and apical actin formation.

*Figure 1—figure supplement 1. I could not find a description of what constitutes a mild vs. severe defect in docking and I worry that this data is also a bit subjective. As this phenotype is likely contributing to many (or all) of the other phenotypes it seems critically important.*

Representative examples as used for classification into “fully docked”, “mild docking defect” and “severe docking defect” are shown in the figures and color coded (white, grey and black boxes shown on lateral projections) in Figure 1 and Figure 1—figure supplement 2. In the revised manuscript, we have added a description to the figure legend to clarify that. Additionally, we have taken one of the data sets and re-analyzed it in a more quantitative fashion: Confocal z-stacks from controls, *cp110* morphants, and rescued *cp110* morphants were analyzed for presence of apically localized basal bodies and basal bodies that remained deep in the cytoplasm. Apically localized basal bodies were defined as present in the first (apical) 4 z-sections containing Centrin4-CFP signal (=1.82µm), deep localized basal bodies were defined as Centrin4-CFP signals located below 1.82µm. Next, we used the 3D Objects Counter plugin in ImageJ to quantify basal bodies that remained deep in the cytoplasm. These automatic quantifications were then inspected and corrected in cases where objects other than basal bodies were detected (assigned object count = 0), and where two or more basal bodies were counted as one (assigned object count =2 or more). This analysis confirmed our previous conclusions and the results are depicted in revised Figure 1—figure supplement 2.

*The claim that Cp110 localizes to "cilia forming basal bodies" seems a bit restrictive, as S3G clearly shows CP110 at non-cilia forming centrioles.*

We do not mean our wording in a restrictive sense. In this manuscript we use “Cp110 localizes to cilia forming basal bodies (and rootlets)” to emphasize our focus on Cp110’s role in cilia formation and function. In the revised manuscript we now mention Cp110’s centriolar role more prominently.

*Also, the main claim of this manuscript is that CP110 levels are critical, so the overexpression (and therefore localization) of CP110 might not reflect the endogenous situation in regards to localization. This data, while reasonable should be qualified and more clearly stated.*

In revised Figure 3 we now show endogenous Cp110 localization data from human airway epithelial cells generated by confocal as well as super-resolution imaging. As in the case of the overexpression data, endogenous Cp110 localizes to the base of cilia (Figure 3), adjacent to the basal body (Figure 3) and at the rootlet (Figure 3). This adds to our previous data on endogenous Cp110 localization to cilia-forming basal bodies in human and mouse airway MCCs, which we in part moved to revised Figure 3—figure supplement 1.

Additionally, although the authors made no comment, careful re-examination of previously published work by Tanos et al. (Genes Dev. 2013, 27:163-168) revealed essentially the same Cp110 localization pattern (medium levels adjacent to the distal end of the cilia-forming mother centriole – low levels at the tip of the rootlet) in primary cilia of human RPE-1 cells using antibody staining of endogenous protein in combination with super-resolution imaging (we provided a figure for the reviewers’ consideration).

*"At the apical membrane, GFP-Cp110 localized between Centrin4-CFP and F-actin (Figure 4—figure supplement 1), supporting the idea that Cp110 might facilitate F-actin binding." I have several concerns with this figure. First off, from my reading of the EM literature there is a complex 3D architecture of actin in MCCs. This analysis appears to be done at the level of the lower basal body / upper rootlet and I am unclear what pool of actin is being analyzed. Second the quality of the image together with the somewhat diffuse actin staining does not seem to me to reflex any consistent relationship between CP110 localization and actin.*

We agree with the reviewer in that the figure was difficult to assess. Therefore, we have removed it from the revised manuscript, and instead provide additional triple staining of Centrin4-CFP, GFP-Cp110 and FAK-mKate (Figure 4, Figure 4—figure supplement 1). This triple staining revealed that Cp110 overlaps posteriorly with Centrin, while FAK mainly overlaps with and extends Cp110 at the posterior domain. These data thus further support a role for Cp110 in linking ciliary adhesions to the basal body. Ciliary adhesions, in turn, were shown to directly interact with the actin cytoskeleton in MCCs by FRET analysis (Antoniades et al. 2014, Dev Cell 28, 70-80).

*"We conclude from these experiments that Cp110 is required for normal formation of ciliary adhesion complexes in MCCs, and suggest that loss of FAK, Vinculin and Paxillin from basal bodies and the apical membrane causes the observed defects in basal body transport, docking and alignment, as well as loss of apical actin." The claim that CP110 somehow regulates the adhesion complex seems concerning. If basal bodies have failed to dock then I am not sure what one would expect to find in regards to the adhesion complex. I am concerned that these adhesion complex phenotypes are completely secondary to docking failure and not that FAK is regulating docking.*

Antoniades et al. (2014, Dev Cell 28, 70-80) have shown that ciliary adhesion complexes in MCCs are first required for apical basal body transport and docking. Therefore, ciliary adhesions act upstream of basal body docking to the apical membrane, in addition to their later role in basal body alignment, which takes place after docking and cilia formation. In revised Figure 3—figure supplement 1 we now show that Cp110 localizes to newly formed basal bodies already during stages when apical basal body transport and docking takes place, which supports our conclusion.

Additionally, we present new data on *cp110* MO-dose dependent phenotypes in MCCs (Figure 1—figure supplement 1). Low *cp110* MO concentrations lead to normal apical transport and docking of basal bodies and successful ciliogenesis, while interfering selectively with proper basal body alignment, which depends on ciliary adhesions at the rootlet tip. Lower Cp110 concentrations are found at the rootlet tip (Figure 3’) and loss of ciliary adhesion complexes is more pronounced at this site (Figure 4, Figure 4—figure supplement 2). We further provide evidence that gain of Cp110 increases FAK levels at basal bodies and rootlets – again, in a dose-dependent manner. Collectively, these data strongly support a specific and dose-dependent effect of Cp110 loss on ciliary adhesion formation, independent of basal body docking.

Our conclusions are also in line with the observations by Yadav et al. (Development, 2016, 143, 1491-1501) that in Cp110-/- mice, 5-10% of primary cilia were still formed, and basal bodies docked to ciliary vesicles were found in about the same frequency. These mice should lack Cp110 protein altogether; therefore, presence of Cp110 at the basal body does not seem to be a strict requirement for ciliary vesicle docking (which also takes place at the distal end and not adjacent to the basal body where Cp110 is found in cilia-forming basal bodies) and cilia formation, but rather promote it at the optimal concentrations and sites. For clarification, we extended our discussion on this point in the third paragraph of the Discussion.

*"In MCCs, gfp-cp110DCentral induced strong aggregation and an increased number of basal bodies (Figure 5—figure supplement 2), implicating Cp110 in regulation of basal body biogenesis and separation via the MCC-specific deuterosome pathway." This is most certainly an overstatement. Multinucleated cells have more "stuff" and therefore make more cilia, independent of what is causing the cytokinesis defect.*

We agree with the reviewer in that the increase in basal body numbers is likely a secondary effect due to the presence of supernumerary centrioles. The sentence now reads as follows: “In MCCs, *gfp-cp110△Central* induced strong aggregation and an increased number of basal bodies (Figure 5—figure supplement 2), likely due to the presence of supernumerary centrioles at the onset of deuterosome-mediated centriole amplification.”

*Reviewer #2 (Additional data files and statistical comments):*

*Most of the data is solid. A better description of some of the "subjective" quantifications should be provided. Full images of the IP would help in the interpretation of the rigor of these experiments.*

We now provide a quantification of basal body localization (Figure 1—figure supplement 2). Full images of the IP presented in Figure 4 are now shown in Figure 4—figure supplement 2.

*Reviewer #3:*

1) The localizations shown for GFP-Cp110 are inconsistent in Figure 3: Figure 3 shows GFP-CP110 away from the distal end of the basal body, in contrast to its localization in Figure 3' at the distal end and possibly at the rootlet, and apparently at both centrioles in the mono-ciliated cells. As the main topic of the MS. is the importance of CP110 levels, the concern about promiscuous localization of over-expressed CP110 should be addressed. Where is the endogenous protein?

In both cases, (revised Figure 3’) GFP-Cp110 localizes adjacent to the basal body in a posterior polarized fashion, with some overlap with the Centrin4-CFP. We understand that this is somewhat more difficult to appreciate in Figure 3’ where we increased brightness of the GFP channel to visualize the harder-to-see population of GFP-Cp110 at the rootlet tip (Figure 7). Therefore, we indicated the positions of Centrin4-CFP (yellow circle) and GFP-Cp110 (green dotted circle) in Figure 3’.

Author response image 1.Left panel shows Centrin-CFP (blue) and GFP-Cp110 (green) in images where green channel brightness is low.In this image Cp110 localizes adjacent to the basal body. Right panel shows Centrin-CFP (blue) and GFP-Cp110 (green) in images where green channel brightness is high. In this image we can visualize the low-level localization of GFP-Cp110 to the rootlet tip, which was not visible in the left panel. Bottom panel shows Clamp-RFP staining of the ciliary rootlet (red). Top row, middle panel shows schematic localization of Cp110 relative to the basal body and the rootlet. Green = Cp110, blue = Centrin4, red = Clamp.**DOI:**
http://dx.doi.org/10.7554/eLife.17557.026

In revised Figure 3 we now show endogenous Cp110 localization data from human airway epithelial cells. As in the case of the overexpression data, endogenous Cp110 localizes to the base of cilia (Figure 3), adjacent to the basal body (Figure 3) and at the rootlet (Figure 3). We have also included a schematic overview of the observed Cp110 localization patterns at centrioles and basal bodies (Figure 3—figure supplement 3).

Additionally, careful re-examination of previously published work by Tanos et al. (Genes Dev. 2013, 27:163-168) revealed essentially the same Cp110 localization pattern (medium levels adjacent to the distal end of the cilia-forming mother centriole, low levels at the tip of the rootlet) in primary cilia of human RPE-1 cells using antibody staining of endogenous protein in combination with super-resolution imaging.

*2) Better controls should be shown for the IP experiments in Figure 4. The co-IPs with FAK-GFP and Centrin4-GFP are effectively the negative controls for each other, so they should be included on the same membrane.*

We thank the reviewer for this comment. Indeed, the experiment was performed on the same membrane and also included Flag-IP of a Flag-Cp110FS△CCD1 construct which also interacted with FAK-GFP and Centrin4-GFP. The full membrane scans are now provided in the revised manuscript Figure 4.

*3) It is not clear whether Cp110-FS is actually expressed, or whether this was an error in the deposited sequence. The Xenopus tropicalis CP110 sequence should be verified by RT-PCR and the correct sequence deposited. The section on the Xt sequence should be rewritten accordingly.*

We have no indication that the Cp110-FS clone (accession # BC167469) is representing a real transcript variant, but we rather conclude that it was a cloning artifact during the initial EST clone collection. The NCBI curators have already identified the artifact in the reference sequence clone independently from us. Therefore, the previous reference sequence for *Xenopus tropicalis cp110* was suppressed with the comment: *“This RefSeq was suppressed temporarily based on the calculation that the encoded protein was shorter than proteins from the putative ortholog from human: CCP110 (GeneID:9738)”* (link: http://www.ncbi.nlm.nih.gov/nuccore/194332527).

The missing adenine is also present in *Xenopus laevis cp110* at the same position (not shown). Figure 8 shows an alignment of the region in question including BC167469 (cp110-fs clone), the updated *Xenopus tropicalis cp110* reference sequence (XM_0129708) and genome sequence (v9.0). In all but BC167469 the Adenine at position 1766 is present.

Author response image 2.Alignment of the *cp110* region, where the missing Adenine was identified in the FS-clone.BC167469, *Xenopus tropicalis* genome 9.0 sequence and the current *Xenopus tropicalis cp110* reference sequence (XM_0129708) are shown.**DOI:**
http://dx.doi.org/10.7554/eLife.17557.027

*4) The expression levels of the various Cp110 deletion constructs appear different by GFP fluorescence. An immunoblot should be included to show that the different effects are not due to different expression levels or altered protein stability.*

We agree with the reviewer that there are some differences in effective expression levels between the constructs. We have indicated that in Figure 5—figure supplement 2 (last column) of our manuscript. As requested, we now show an immunoblot of GFP-tagged Cp110 constructs (Figure 5—figure supplement 1), which shows that all constructs are expressed in relevant stages (stage 20) at similar levels, with GFP-Cp110△Central (which lacks the RXL and KEN motifs) and GFP-Cp110△C (which lacks the *miR-34/449* target sites in its mRNA) showing the highest expression. Most importantly, GFP-Cp110△CCD1&2 (which shows low cilia inhibition potential) is not significantly less expressed than constructs that inhibit cilia efficiently.

*5) Arising from point 4., are post-translational regulation of Cp110 levels important in MCCs? Given the published work on the cell cycle regulation of Cp110, some discussion/ clarification of this point for the differentiated cells would be relevant.*

We agree with the reviewer that post-translational mechanisms likely contribute to the regulation of Cp110 levels in all cells, including MCCs to some extent. We indicate that by stating: “Interestingly, the negative effects on basal bodies and cell size were variable among cilia-inhibiting constructs (Figure 5—figure supplement 1): Most prominently, deletion of a central domain (GFP-Cp110ΔCentral) containing most phosphorylation sites, RXL and KEN domains (proteasome targeting motifs), induced much stronger effects than full-length Cp110 (Figure 5; Figure 5—figure supplement 1). This suggests that central domain deletion generated a hypermorphic protein which was released from negative regulation by the proteasomal machinery.”

[Editors' note: further revisions were requested prior to acceptance, as described below.]

*The main concern remains that the conclusions on the localization of Cp110 are largely based on expression of GFP-CP110, which may be over-expressed relative to endogenous protein, and that they lack quantitation. The authors should present evidence that the unconventional localization is not background and that it is a statistically significant occurrence. It was recognized that the findings of the paper are novel and interesting and should not be held hostage to the lack of all the necessary reagents to confirm native localization and interactions. However, the reviewers felt that the claims of the paper on these issues should be modified and the caveats made clearer.*

We thank the reviewers and editors for the constructive criticism on our manuscript, which helped to improve it. However, we are puzzled about the above comment, that “The main concern remains that the conclusions on the localization of Cp110 are largely based on expression of GFP-CP110, which may be over-expressed relative to endogenous protein, and that they lack quantitation. The authors should present evidence that the unconventional localization is not background and that it is a statistically significant occurrence.”

Indeed, we added further immunostaining (conventional and super-resolution) of endogenous proteins to the paper, which addressed this point directly. The staining is reproducible, and we provide the number of experiments/donors and cells in the figure legends to figures Figure 3 donors = 1, n MCCs =4), 3D (n donors =1, n MCCs = 12), 3E-F (n donors = 1, n MCCs = 3 each for confocal and 3D-SIM), Figure 3—figure supplement 1 (mice, n=4; and including a control staining with only secondary antibody). In all species and cases, Cp110 was found at cilia-forming basal bodies. Furthermore, with respect to the possibility that GFP might cause background, other papers have demonstrated that expression of GFP alone does not result in profound basal body localization (including Antoniades, I., Stylianou, P. & Skourides, P. A. Making the connection: ciliary adhesion complexes anchor Basal bodies to the actin cytoskeleton. Dev. Cell28,70–80 (2014).).

We have however, further addressed the reviewer’s concern in our re-revised manuscript by including additional data on the localization of an overexpressed, GFP-tagged Cep97 construct in the *Xenopus* epidermis and present these findings in revised Figure 3—figure supplement 3 as well as in the third paragraph of the subsection “Cp110 localizes to cilia-forming basal bodies”. Cep97 cooperates with Cp110 in cilia suppression at centrosomes (Spektor, A., Tsang, W. Y., Khoo, D. & Dynlacht, B. D. Cep97 and CP110 suppress a cilia assembly program. Cell130, 678–90 (2007)). We now show that while GFP- Cep97 localizes robustly to centrosomes, it does not localize to MCC basal bodies and does not suppress cilia formation, which further supports our conclusion that GFP-Cp110 localization adjacent to basal bodies and to rootlet tips are specific. We hope that this addresses the reviewer’s remaining concerns.

We are unsure what kind of “statistical significance” the reviewer(s) expect from us since all our staining experiments reliably detected Cp110 at basal bodies?